# Modular Meta-Learning with Shrinkage

**Yutian Chen**[*]      **Abram L. Friesen**[*]      **Feryal Behbahani**      **Arnaud Doucet**

**David Budden**          **Matthew W. Hoffman**          **Nando de Freitas**

DeepMind
London, UK
`{yutianc, abef}@google.com`

## Abstract

Many real-world problems, including multi-speaker text-to-speech synthesis, can greatly benefit from the ability to meta-learn large models with only a few task-specific components. Updating only these task-specific modules then allows the model to be adapted to low-data tasks for as many steps as necessary without risking overfitting. Unfortunately, existing meta-learning methods either do not scale to long adaptation or else rely on handcrafted task-specific architectures. Here, we propose a meta-learning approach that obviates the need for this often sub-optimal hand-selection. In particular, we develop general techniques based on Bayesian shrinkage to automatically discover and learn both task-specific and general reusable modules. Empirically, we demonstrate that our method discovers a small set of meaningful task-specific modules and outperforms existing meta-learning approaches in domains like few-shot text-to-speech that have little task data and long adaptation horizons. We also show that existing meta-learning methods including MAML, iMAML, and Reptile emerge as special cases of our method.

## 1   Introduction

The goal of meta-learning is to extract shared knowledge from a large set of training tasks to solve held-out tasks more efficiently. One avenue for achieving this is to learn task-agnostic modules and reuse or repurpose these for new tasks. Reusing or repurposing modules can reduce overfitting in low-data regimes, improve interpretability, and facilitate the deployment of large multi-task models on limited-resource devices as parameter sharing allows for significant savings in memory. It can also enable batch evaluation of reused modules across tasks, which can speed up inference time on GPUs.

These considerations are important in domains like few-shot text-to-speech synthesis (TTS), characterized by large speaker-adaptable models, limited training data for speaker adaptation, and long adaptation horizons [1]. Adapting the model to a new task for more optimization steps generally improves the model capacity without increasing the number of parameters. However, many meta-learning methods are designed for quick adaptation, and hence are inapplicable in this *few data and long adaptation* regime. For those that are applicable [2–5], adapting the full model to few data can then fail because of overfitting. To overcome this, modern TTS models combine shared core modules with handcrafted, adaptable, speaker-specific modules [6, 1, 7, 8]. This hard coding strategy is often suboptimal. As data increases, these hard-coded modules quickly become a bottleneck for further improvement, even in a few-shot regime. For this reason, we would like to automatically learn the smallest set of modules needed to adapt to a new speaker and then adapt these for as long as needed.

Automatically learning reusable and broadly applicable modular mechanisms is an open challenge in

---

[*]Equal contribution.

causality, transfer learning, and domain adaptation [9–12]. In meta-learning, most existing gradient-based algorithms, such as MAML [13], do not encourage meta-training to develop reusable and general modules, and either ignore reusability or manually choose the modules to fix [14, 15, 5, 16–18]. Some methods implicitly learn a simple form of modularity for some datasets [17, 19] but it is limited.

In this paper, we introduce a novel approach for automatically finding reusable modules. Our approach employs a principled hierarchical Bayesian model that exploits a statistical property known as shrinkage, meaning that low-evidence estimates tend towards their prior mean; e.g., see Gelman et al. [20]. This is accomplished by first partitioning any neural network into arbitrary groups of parameters, which we refer to as modules. We assign a Gaussian prior to each module with a scalar variance. When the variance parameter shrinks to zero for a specific module, as it does if the data does not require the module parameters to deviate from the prior mean, then all of the module's parameters become tied to the prior mean during task adaptation. This results in a set of automatically learned modules that can be reused at deployment time and a sparse set of remaining modules that are adapted subject to the estimated prior.

Estimating the prior parameters in our model corresponds to meta-learning, and we present two principled methods for this based on maximizing the predictive log-likelihood. Importantly, both methods allow many adaptation steps. By considering non-modular variants of our model, we show that MAML [13], Reptile [2], and iMAML [3] emerge as special cases. We compare our proposed shrinkage-based methods with their non-modular baselines on multiple low-data, long-adaptation domains, including a challenging variant of Omniglot and TTS. Our modular, shrinkage-based methods exhibit higher predictive power in low-data regimes without sacrificing performance when more data is available. Further, the discovered modular structures corroborate common knowledge about network structure in computer vision and provide new insights about WaveNet [21] layers in TTS.

In summary, we introduce a hierarchical Bayesian model for modular meta-learning along with two parameter-estimation methods, which we show generalize existing meta-learning algorithms. We then demonstrate that our approach enables identification of a small set of meaningful task-specific modules. Finally, we show that our method prevents overfitting and improves predictive performance on problems that require many adaptation steps given only small amounts of data.

## 1.1 Related Work

Multiple Bayesian meta-learning approaches have been proposed to either provide model uncertainty in few-shot learning [22–25] or to provide a probabilistic interpretation and extend existing non-Bayesian works [26–28]. However, to the best of our knowledge, none of these account for modular structure in their formulation. While we use point estimates of variables for computational reasons, more sophisticated inference methods from these works can also be used within our framework.

Modular meta-learning approaches based on MAML-style backpropagation through short task adaptation horizons have also been proposed. The most relevant of these, Alet et al. [29], proposes to learn a modular network architecture, whereas our work identifies the adaptability of each module. In other work, Zintgraf et al. [16] hand-designs the task-specific and shared parameters, and the M-Net in Lee and Choi [14] provides an alternative method for learning adaptable modules by sampling binary mask variables. In all of the above, however, backpropagating through task adaptation is computationally prohibitive when applied to problems that require longer adaptation horizons. While it is worth investigating how to extend these works to this setting, we leave this for future work.

Many other meta-learning works [14, 15, 5, 30–32] learn the learning rate (or a preconditioning matrix) which can have a similar modular regularization effect as our approach if applied modularly with fixed adaptation steps. However, these approaches and ours pursue fundamentally different and orthogonal goals. Learning the learning rate aims at fast optimization by fitting to local curvature, without changing the task loss or associated stationary point. In contrast, our method learns to control how far each module will move from its initialization by changing that stationary point. In problems requiring long task adaptation, these two approaches lead to different behaviors, as we demonstrate in Appendix C. Further, most of these works also rely on backpropagating through gradients, which does not scale well to the long adaptation horizons considered in this work. Overall, generalizing beyond the training horizon is challenging for "learning-to-optimize" approaches [33, 34]. While WarpGrad [5] does allow long adaptation horizons, it is not straightforward to apply its use of a functional mapping from inputs to a preconditioning matrix for module discovery and we leave this for future work.

**Algorithm 1:** Meta-learning pseudocode.

**Input:** Batch size $B$, steps $L$, and learning rate $\alpha$
Initialize $\phi$
**while** *not done* **do**
    $\{t_1, \ldots, t_B\} \leftarrow$ sample mini-batch of tasks
    **for** *each task $t$ in $\{t_1, \ldots, t_B\}$* **do**
        Initialize $\boldsymbol{\theta}_t \leftarrow \phi$
        **for** *step $l = 1 \ldots L$* **do**
            $\boldsymbol{\theta}_t \leftarrow \text{TASKADAPT}(\mathcal{D}_t, \phi, \boldsymbol{\theta}_t)$
        **end**
    **end**
    *// Meta update*
    $\phi \leftarrow \phi - \alpha \cdot \frac{1}{B} \sum_t \Delta_t(\mathcal{D}_t, \phi, \boldsymbol{\theta}_t)$
**end**

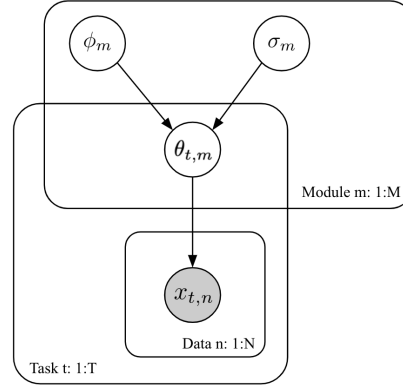

Figure 1: (Left) Structure of a typical meta-learning algorithm. (Right) Bayesian shrinkage graphical model. The shared meta parameters $\phi$ serve as the initialization of the neural network parameters for each task $\boldsymbol{\theta}_t$. The $\boldsymbol{\sigma}$ are shrinkage parameters. By learning these, the model automatically decides which subsets of parameters (i.e., modules) to fix for all tasks and which to adapt at test time.

Finally, L2 regularization is also used for task adaptation in continual learning [35, 36] and meta-learning [37–39, 3]. However, in these works, the regularization scale(s) are either treated as hyper-parameters or adapted per dimension with a different criterion from our approach. It is not clear how to learn the regularization scale at a module level in these works.

## 2 Gradient-based Meta-Learning

We begin with a general overview of gradient-based meta-learning, as this is one of the most common approaches to meta-learning. In this regime, we assume that there are many tasks, indexed by $t$, and that each of these tasks has few data. That is, each task is associated with a finite dataset $\mathcal{D}_t = \{\mathbf{x}_{t,n}\}$ of size $N_t$, which can be partitioned into training and validation sets, $\mathcal{D}_t^{\text{train}}$ and $\mathcal{D}_t^{\text{val}}$ respectively. To solve a task, gradient-based meta-learning adapts *task-specific parameters* $\boldsymbol{\theta}_t \in \mathbb{R}^D$ by minimizing a loss function $\ell(\mathcal{D}_t; \boldsymbol{\theta}_t)$ using a local optimizer. Adaptation is made more efficient by sharing a set of *meta parameters* $\phi \in \mathbb{R}^D$ between tasks, which are typically used to initialize the task parameters.

Algorithm 1 summarizes a typical stochastic meta-training procedure, which includes MAML [13], implicit MAML (iMAML) [3], and Reptile [2]. Here, TASKADAPT executes one step of optimization of the task parameters. The meta-update $\Delta_t$ specifies the contribution of task $t$ to the meta parameters. At test time, multiple steps of TASKADAPT are run on each new test task.

MAML implements task adaptation by applying gradient descent to minimize the training loss $\ell_t^{\text{train}}(\boldsymbol{\theta}_t) = \ell(\mathcal{D}_t^{\text{train}}; \boldsymbol{\theta}_t)$ with respect to the task parameters. It then updates the meta parameters by gradient descent on the validation loss $\ell_t^{\text{val}}(\boldsymbol{\theta}_t) = \ell(\mathcal{D}_t^{\text{val}}; \boldsymbol{\theta}_t)$, resulting in the meta update $\Delta_t^{\text{MAML}} = \nabla_{\phi} \ell_t^{\text{val}}(\boldsymbol{\theta}_t(\phi))$. This approach treats the task parameters as a function of the meta parameters, and hence requires backpropagating through the entire $L$-step task adaptation process. When $L$ is large, as in TTS systems [1], this becomes computationally prohibitive.

Reptile and iMAML avoid this computational burden of MAML. Reptile instead optimizes $\boldsymbol{\theta}_t$ on the entire dataset $\mathcal{D}_t$, and moves $\phi$ towards the adapted task parameters, yielding $\Delta_t^{\text{Reptile}} = \phi - \boldsymbol{\theta}_t$. Conversely, iMAML introduces an L2 regularizer $\frac{\lambda}{2}||\boldsymbol{\theta}_t - \phi||^2$ and optimizes the task parameters on the regularized training loss. Provided that this task adaptation process converges to a stationary point, *implicit differentiation* enables the computation of the meta gradient based only on the final solution of the adaptation process, $\Delta_t^{\text{iMAML}} = \left(\mathbf{I} + \frac{1}{\lambda}\nabla_{\boldsymbol{\theta}_t}^2 \ell_t^{\text{train}}(\boldsymbol{\theta}_t)\right)^{-1} \nabla_{\boldsymbol{\theta}_t} \ell_t^{\text{val}}(\boldsymbol{\theta}_t)$. See Rajeswaran et al. [3] for details.

## 3 Modular Bayesian Meta-Learning

In standard meta-learning, the meta parameters $\phi$ provide an initialization for the task parameters $\boldsymbol{\theta}$ at test time. That is, all the neural network parameters are treated equally, and hence they must all be updated at test time. This strategy is inefficient and prone to overfitting. To overcome it, researchers

often split the network parameters into two groups, a group that varies across tasks and a group that is shared; see for example [1, 16]. This division is heuristic, so in this work we explore ways of automating it to achieve better results and to enable automatic discovery of task independent modules. More precisely, we assume that the network parameters can be partitioned into $M$ disjoint modules $\boldsymbol{\theta}_t = (\boldsymbol{\theta}_{t,1}, \ldots, \boldsymbol{\theta}_{t,m}, \ldots, \boldsymbol{\theta}_{t,M})$ where $\boldsymbol{\theta}_{t,m}$ are the parameters in module $m$ for task $t$. This view of modules is very general. Modules can correspond to layers, receptive fields, the encoder and decoder in an auto-encoder, the heads in a multi-task learning model, or any other grouping of interest.

We adopt a hierarchical Bayesian model, shown in Figure 1, with a factored probability density:

$$p(\boldsymbol{\theta}_{1:T}, \mathcal{D}|\boldsymbol{\sigma}^2, \boldsymbol{\phi}) = \prod_{t=1}^{T}\prod_{m=1}^{M} \mathcal{N}(\boldsymbol{\theta}_{t,m}|\boldsymbol{\phi}_m, \sigma_m^2\mathbf{I}) \prod_{t=1}^{T} p(\mathcal{D}_t|\boldsymbol{\theta}_t). \tag{1}$$

The $\boldsymbol{\theta}_{t,m}$ are conditionally independent and normally distributed $\boldsymbol{\theta}_{t,m} \sim \mathcal{N}(\boldsymbol{\theta}_{t,m}|\boldsymbol{\phi}_m, \sigma_m^2\mathbf{I})$ with mean $\boldsymbol{\phi}_m$ and variance $\sigma_m^2$, where $\mathbf{I}$ is the identity matrix.

The $m$-th module scalar shrinkage parameter $\sigma_m^2$ measures the degree to which $\boldsymbol{\theta}_{t,m}$ can deviate from $\boldsymbol{\phi}_m$. More precisely, for values of $\sigma_m^2$ near zero, the difference between parameters $\boldsymbol{\theta}_{t,m}$ and mean $\boldsymbol{\phi}_m$ will be shrunk to zero and thus module $m$ will become task independent. Thus by learning the parameters $\boldsymbol{\sigma}^2$, we discover which modules are task independent. These independent modules can be reused at test time, reducing the computational burden of adaptation and likely improving generalization. Shrinkage is often used in automatic relevance determination for sparse feature selection [40].

We place uninformative priors on $\boldsymbol{\phi}_m$ and $\sigma_m$, and follow an empirical Bayes approach to learn their values from data. This formulation allows the model to automatically learn which modules to reuse—i.e. those modules for which $\sigma_m^2$ is near zero—and which to adapt at test time.

## 4 Meta-Learning as Parameter Estimation

By adopting the hierarchical Bayesian model from the previous section, the problem of meta-learning becomes one of estimating the parameters $\boldsymbol{\phi}$ and $\boldsymbol{\sigma}^2$. A standard solution to this problem is to maximize the marginal likelihood $p(\mathcal{D}|\boldsymbol{\phi}, \boldsymbol{\sigma}^2) = \int p(\mathcal{D}|\boldsymbol{\theta})p(\boldsymbol{\theta}|\boldsymbol{\phi}, \boldsymbol{\sigma}^2)\,\mathrm{d}\boldsymbol{\theta}$. We can also assign a prior over $\boldsymbol{\phi}$. In both cases, the marginalizations are intractable, so we must seek scalable approximations.

It may be tempting to estimate the parameters by maximizing $p(\boldsymbol{\theta}_{1:T}, \mathcal{D}|\boldsymbol{\sigma}^2, \boldsymbol{\phi})$ w.r.t. $(\boldsymbol{\sigma}^2, \boldsymbol{\phi}, \boldsymbol{\theta}_{1:T})$, but the following lemma suggests that this naive approach leads to all task parameters being tied to the prior mean, i.e. no adaptation will occur (see Appendix A for a proof):

**Lemma 1.** *The function $f : (\boldsymbol{\sigma}^2, \boldsymbol{\phi}, \boldsymbol{\theta}_{1:T}) \mapsto \log p(\boldsymbol{\theta}_{1:T}, \mathcal{D}|\boldsymbol{\sigma}^2, \boldsymbol{\phi})$ diverges to $+\infty$ as $\boldsymbol{\sigma}^2 \to 0^+$ when $\boldsymbol{\theta}_{t,m} = \boldsymbol{\phi}_m$ for all $t \in \{1, ..., T\}, m \in \{1, ..., M\}$.*

The undesirable result of Lemma 1 is caused by the use of point estimate of $\boldsymbol{\theta}_{1:T}$ in maximum likelihood estimation of $\boldsymbol{\sigma}^2$ and $\boldsymbol{\phi}$. In the following two subsections, we propose two principled alternative approaches for parameter estimation based on maximizing the predictive likelihood over validation subsets.

### 4.1 Parameter estimation via the predictive likelihood

Our goal is to minimize the average negative predictive log-likelihood over $T$ validation tasks,

$$\ell_{\mathrm{PLL}}(\boldsymbol{\sigma}^2, \boldsymbol{\phi}) = -\frac{1}{T}\sum_{t=1}^{T} \log p\left(\mathcal{D}_t^{\mathrm{val}}|\mathcal{D}_t^{\mathrm{train}}, \boldsymbol{\sigma}^2, \boldsymbol{\phi}\right) = \frac{1}{T}\sum_{t=1}^{T} \log \int p(\mathcal{D}_t^{\mathrm{val}}|\boldsymbol{\theta}_t)\, p(\boldsymbol{\theta}_t|\mathcal{D}_t^{\mathrm{train}}, \boldsymbol{\sigma}^2, \boldsymbol{\phi})\,\mathrm{d}\boldsymbol{\theta}_t. \tag{2}$$

To justify this goal, assume that the training and validation data is distributed i.i.d according to some distribution $\nu(\mathcal{D}_t^{\mathrm{train}}, \mathcal{D}_t^{\mathrm{val}})$. Then, the law of large numbers implies that as $T \to \infty$,

$$\ell_{\mathrm{PLL}}(\boldsymbol{\sigma}^2, \boldsymbol{\phi}) \to \mathbb{E}_{\nu(\mathcal{D}_t^{\mathrm{train}})}[\mathrm{KL}(\nu(\mathcal{D}_t^{\mathrm{val}}|\mathcal{D}_t^{\mathrm{train}})||p\left(\mathcal{D}_t^{\mathrm{val}}|\mathcal{D}_t^{\mathrm{train}}, \boldsymbol{\sigma}^2, \boldsymbol{\phi}\right))] + \mathrm{H}(\nu(\mathcal{D}_t^{\mathrm{val}}|\mathcal{D}_t^{\mathrm{train}})), \tag{3}$$

where KL denotes the Kullback-Leibler divergence and H the entropy. Thus minimizing $\ell_{\mathrm{PLL}}$ with respect to the meta parameters corresponds to selecting the predictive distribution $p\left(\mathcal{D}_t^{\mathrm{val}}|\mathcal{D}_t^{\mathrm{train}}, \boldsymbol{\sigma}^2, \boldsymbol{\phi}\right)$ that is closest (approximately) to the true predictive distribution $\nu(\mathcal{D}_t^{\mathrm{val}}|\mathcal{D}_t^{\mathrm{train}})$ on average. This criterion can be thought of as an approximation to a Bayesian cross-validation criterion [41].

Computing $\ell_{\text{PLL}}$ is not feasible due to the intractable integral in equation (2). Instead we make use of a simple maximum a posteriori (MAP) approximation of the task parameters:

$$\hat{\boldsymbol{\theta}}_t(\boldsymbol{\sigma}^2, \boldsymbol{\phi}) = \underset{\boldsymbol{\theta}_t}{\arg\min} \, \ell_t^{\text{train}}(\boldsymbol{\theta}_t, \boldsymbol{\sigma}^2, \boldsymbol{\phi}), \text{ where } \ell_t^{\text{train}} := -\log p\left(\mathcal{D}_t^{\text{train}} | \boldsymbol{\theta}_t\right) - \log p\left(\boldsymbol{\theta}_t | \boldsymbol{\sigma}^2, \boldsymbol{\phi}\right). \quad (4)$$

We note for clarity that $\ell_t^{\text{train}}$ corresponds to the negative log of equation (1) for a single task. Using these MAP estimates, we can approximate $\ell_{\text{PLL}}(\boldsymbol{\sigma}^2, \boldsymbol{\phi})$ as follows:

$$\hat{\ell}_{\text{PLL}}(\boldsymbol{\sigma}^2, \boldsymbol{\phi}) = \frac{1}{T} \sum_{t=1}^{T} \ell_t^{\text{val}}(\hat{\boldsymbol{\theta}}_t(\boldsymbol{\sigma}^2, \boldsymbol{\phi})), \quad \text{where } \ell_t^{\text{val}} := -\log p\left(\mathcal{D}_t^{\text{val}} | \hat{\boldsymbol{\theta}}_t\right). \quad (5)$$

We use this loss to construct a meta-learning algorithm with the same structure as Algorithm 1. Individual task adaptation follows from equation (4) and meta updating from minimizing $\hat{\ell}_{\text{PLL}}(\boldsymbol{\sigma}^2, \boldsymbol{\phi})$ in equation (5) with an unbiased gradient estimator and a mini-batch of sampled tasks.

Minimizing equation (5) is a bi-level optimization problem that requires solving equation (4) implicitly. If optimizing $\ell_t^{\text{train}}$ requires only a small number of local optimization steps, we can compute the update for $\boldsymbol{\phi}$ and $\boldsymbol{\sigma}^2$ with back-propagation through $\hat{\boldsymbol{\theta}}_t$, yielding

$$\Delta_t^{\boldsymbol{\sigma}\text{-MAML}} = \nabla_{\boldsymbol{\sigma}^2, \boldsymbol{\phi}} \, \ell_t^{\text{val}}(\hat{\boldsymbol{\theta}}_t(\boldsymbol{\sigma}^2, \boldsymbol{\phi})). \quad (6)$$

This update reduces to that of MAML if $\sigma_m^2 \to \infty$ for all modules and is thus denoted as $\boldsymbol{\sigma}$-MAML.

We are however more interested in long adaptation horizons for which back-propagation through the adaptation becomes computationally expensive and numerically unstable. Instead, we apply the *implicit function theorem* on equation (4) to compute the gradient of $\hat{\boldsymbol{\theta}}_t$ with respect to $\boldsymbol{\sigma}^2$ and $\boldsymbol{\phi}$, giving

$$\Delta_t^{\boldsymbol{\sigma}\text{-iMAML}} = -\nabla_{\boldsymbol{\theta}_t} \ell_t^{\text{val}}(\boldsymbol{\theta}_t) \mathbf{H}_{\boldsymbol{\theta}_t \boldsymbol{\theta}_t}^{-1} \mathbf{H}_{\boldsymbol{\theta}_t \boldsymbol{\Phi}}, \quad (7)$$

where $\boldsymbol{\Phi} = (\boldsymbol{\sigma}^2, \boldsymbol{\phi})$ is the full vector of meta-parameters, $\mathbf{H}_{ab} = \nabla_{a,b}^2 \ell_t^{\text{train}}$, and derivatives are evaluated at the stationary point $\boldsymbol{\theta}_t = \hat{\boldsymbol{\theta}}_t(\boldsymbol{\sigma}^2, \boldsymbol{\phi})$. A detailed derivation is provided in Appendix B.1. Various approaches have been proposed to approximate the inverse Hessian [3, 42–44]. We use the conjugate gradient algorithm. We show in Appendix B.2 that the meta update for $\boldsymbol{\phi}$ is equivalent to that of iMAML when $\sigma_m^2$ is constant for all $m$, and thus refer to this more general method as $\boldsymbol{\sigma}$-iMAML.

In summary, our goal of maximizing the predictive likelihood of validation data conditioned on training data for our hierarchical Bayesian model results in modular generalizations of the MAML and iMAML approaches. In the following subsection, we will see that the same is also true for Reptile.

## 4.2 Estimating $\phi$ via MAP approximation

If we instead let $\boldsymbol{\phi}$ be a random variable with prior distribution $p(\boldsymbol{\phi})$, then we can derive a different variant of the above algorithm. We return to the predictive log-likelihood introduced in the previous section but now integrate out the central parameter $\boldsymbol{\phi}$. Since $\boldsymbol{\phi}$ depends on all training data we will rewrite the average predictive likelihood in terms of the joint posterior over $(\boldsymbol{\theta}_{1:T}, \boldsymbol{\phi})$, i.e.

$$\ell_{\text{PLL}}(\boldsymbol{\sigma}^2) = -\frac{1}{T} \log p(\mathcal{D}_{1:T}^{\text{val}} | \mathcal{D}_{1:T}^{\text{train}}, \boldsymbol{\sigma}^2) = -\frac{1}{T} \log \int \left(\prod_{t=1}^{T} p(\mathcal{D}_t^{\text{val}} | \boldsymbol{\theta}_t)\right) p(\boldsymbol{\theta}_{1:T}, \boldsymbol{\phi} | \mathcal{D}_{1:T}^{\text{train}}, \boldsymbol{\sigma}^2) \mathrm{d}\boldsymbol{\theta}_{1:T} \mathrm{d}\boldsymbol{\phi}. \quad (8)$$

Again, to address the intractable integral we make use of a MAP approximation, except in this case we approximate both the task parameters and the central meta parameter as

$$(\hat{\boldsymbol{\theta}}_{1:T}(\boldsymbol{\sigma}^2), \hat{\boldsymbol{\phi}}(\boldsymbol{\sigma}^2)) = \underset{\boldsymbol{\theta}_{1:T}, \boldsymbol{\phi}}{\arg\min} -\log p(\boldsymbol{\theta}_{1:T}, \boldsymbol{\phi} | \mathcal{D}_{1:T}^{\text{train}}, \boldsymbol{\sigma}^2) = \underset{\boldsymbol{\theta}_{1:T}, \boldsymbol{\phi}}{\arg\min} \sum_{t=1}^{T} \ell_t^{\text{train}}(\boldsymbol{\theta}_t, \boldsymbol{\sigma}^2, \boldsymbol{\phi}) - \log p(\boldsymbol{\phi}). \quad (9)$$

We assume a flat prior for $\boldsymbol{\phi}$ in this paper and thus drop the second term above. Note that when the number of training tasks $T$ is small, an informative prior would be preferred. We can then estimate

| | | Fixed $\boldsymbol{\sigma}^2$ | Learned $\boldsymbol{\sigma}^2$ | Allows long adaptation? |
|---|---|---|---|---|
| $\hat{\boldsymbol{\phi}}_{\text{joint}}$ | | Reptile | $\boldsymbol{\sigma}$-Reptile | ✓ |
| $\hat{\boldsymbol{\phi}}_{\text{PLL}}$ | Back-prop. | MAML | $\boldsymbol{\sigma}$-MAML | × |
| | Implicit grad. | iMAML | $\boldsymbol{\sigma}$-iMAML | ✓ |

Table 1: The above algorithms result from different approximations to the predictive likelihood.

the shrinkage parameter $\boldsymbol{\sigma}^2$ by plugging this approximation into Eq. (8). This method gives the same task adaptation update as the previous section but a meta update of the form

$$\Delta_{t,\boldsymbol{\phi}_m}^{\boldsymbol{\sigma}\text{-Reptile}} = \frac{1}{\sigma_m^2}(\boldsymbol{\phi}_m - \hat{\boldsymbol{\theta}}_{m,t}), \;\; \Delta_{t,\boldsymbol{\sigma}^2}^{\boldsymbol{\sigma}\text{-Reptile}} = -\nabla_{\boldsymbol{\theta}_t}\ell_t^{\text{val}}(\boldsymbol{\theta}_t)\mathbf{H}_{\boldsymbol{\theta}_t\boldsymbol{\theta}_t}^{-1}\mathbf{H}_{\boldsymbol{\theta}_t\boldsymbol{\sigma}^2}\,, \tag{10}$$

where derivatives are evaluated at $\boldsymbol{\theta}_t = \hat{\boldsymbol{\theta}}_t(\boldsymbol{\sigma}^2)$, and the gradient of $\hat{\boldsymbol{\phi}}$ w.r.t. $\boldsymbol{\sigma}^2$ is ignored. Due to lack of space, further justification and derivation of this approach is provided in Appendices B.3 and B.4.

We can see that Reptile is a special case of this method when $\sigma_m^2 \to \infty$ and we choose a learning rate proportional to $\sigma_m^2$ for $\boldsymbol{\phi}_m$. We thus refer to it as $\boldsymbol{\sigma}$-Reptile.

Table 1 compares our proposed algorithms with existing algorithms in the literature. Our three algorithms reduce to the algorithms on the left when $\sigma_m^2 \to \infty$ or a constant scalar for all modules. Another variant of MAML for long adaptation, first-order MAML, can be recovered as a special case of iMAML when using one step of conjugate gradient descent to compute the inverse Hessian [3].

### 4.3 Task-Specific Module Selection

When the parameters $\boldsymbol{\phi}$ and $\boldsymbol{\sigma}^2$ are estimated accurately, the values of $\sigma_m^2$ for task-independent modules shrink towards zero. The remaining modules with non-zero $\sigma_m^2$ are considered task-specific [40]. In practice, however, the estimated value of $\sigma_m^2$ will never be exactly zero due to the use of stochastic optimization and the approximation in estimating the meta gradients. We therefore apply a weak regularization on $\boldsymbol{\sigma}^2$ to encourage its components to be small unless nonzero values are supported by evidence from the task data (see Appendix D for details).

While the learned $\sigma_m^2$ values are still non-zero, in most of our experiments below, the estimated value $\sigma_m^2$ for task-independent modules is either numerically indistinguishable from 0 or at least two orders of magnitude smaller than the value for the task-specific modules. This reduces the gap between meta-training, where all modules have non-zero $\sigma_m^2$, and meta-testing, where a sparse set of modules are selected for adaptation. The exception to this is the text-to-speech experiment where we find the gap of $\sigma_m^2$ between modules to not be as large as in the image experiments. Thus, we instead rank the modules by the value of $\sigma_m^2$ and select the top-ranked modules as task-specific.

Ranking and selecting task-specific modules using the estimated value of $\sigma_m^2$ allows us to trade off between module sparsity and model capacity in practice, and achieves robustness against overfitting. It remains unclear in theory, however, if the task sensitivity of a module is always positively correlated with $\sigma_m^2$ especially when the size of modules varies widely. This is an interesting question that we leave for future investigation.

## 5 Experimental Evaluation

We evaluate our shrinkage-based methods on challenging meta-learning domains that have small amounts of data and require long adaptation horizons, such as few-shot text-to-speech voice synthesis. The aim of our evaluation is to answer the following three questions: (1) Does shrinkage enable automatic discovery of a small set of task-specific modules? (2) Can we adapt only the task-specific modules without sacrificing performance? (3) Does incorporating a shrinkage prior improve performance and robustness to overfitting in problems with little data and long adaptation horizons?

### 5.1 Experiment setup

Our focus is on long adaptation, low data regimes. To this end, we compare iMAML and Reptile to their corresponding shrinkage variants, $\boldsymbol{\sigma}$-iMAML and $\boldsymbol{\sigma}$-Reptile. For task adaptation with the

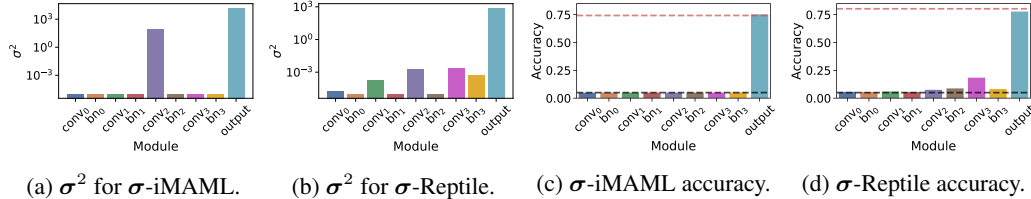

(a) $\sigma^2$ for $\sigma$-iMAML.  (b) $\sigma^2$ for $\sigma$-Reptile.  (c) $\sigma$-iMAML accuracy.  (d) $\sigma$-Reptile accuracy.

Figure 2: Module discovery with $\sigma$-iMAML and $\sigma$-Reptile for large-data augmented Omniglot. (a,b) The learned $\sigma^2$ for each module (y-axis is log scale). (c,d) Test accuracy when only that layer is adapted versus when all layers are adapted using the learned $\sigma^2$.

shrinkage variants, we use proximal gradient descent [45] for the image experiments, and introduce a proximal variant of Adam [46] (pseudocode in Algorithm 3) for the text-to-speech (TTS) experiments. The proximal methods provide robustness to changes in the prior strength $\sigma^2$ over time. We provide further algorithmic details in Appendix D. We evaluate on the following domains.

**Few-shot image classification.** We use the augmented Omniglot protocol of Flennerhag et al. [4], which necessitates long-horizon adaptation. For each alphabet, 20 characters are sampled to define a 20-class classification problem. The domain is challenging because both train and test images are randomly augmented. Following Flennerhag et al. [4], we use a 4-layer convnet and perform 100 steps of task adaptation. We consider two regimes: (**Large-data regime**) We use 30 training alphabets ($T = 30$), 15 training images ($K = 15$), and 5 validation images per class. Each image is randomly re-scaled, translated, and rotated. (**Small-data regime**) To study the effect of overfitting, we vary $T \in \{5, 10, 15, 20\}$ and $K \in \{1, 3, 5, 10, 15\}$, and augment only by scaling and translating.

**Text-to-speech voice synthesis.** Training a neural TTS model from scratch typically requires tens of hours of speech data. In the few-shot learning setting [6, 1, 7, 8], the goal is to adapt a trained model to a new speaker based on only a few minutes of data. Earlier work unsuccessfully applied fast-adaptation methods such as MAML to synthesizing utterances [1]. Instead, their state-of-the-art method first pretrains a multi-speaker model comprised of a shared core network and a speaker embedding module and then finetunes either the entire model or the embedding only. We remove the manually-designed speaker embedding layers and perform task adaptation and meta-updates on only the core network.

The core network is a WaveNet vocoder model [21] with 30 residual causal dilated convolutional blocks as the backbone, consisting of roughly 3M parameters. For computational reasons, we use only one quarter of the channels of the standard network. As a result, sample quality does not reach production level but we expect the comparative results to apply to the full network. We meta-train with tasks of 100 training utterances (about 8 minutes) using 100 task adaptation steps, then evaluate on held-out speakers with either 100 or 50 (about 4 minutes) utterances and up to 10,000 adaptation steps. For more details see Appendix E.2.

**Short adaptation.** While our focus is on long adaptation, we conduct experiments on short-adaptation datasets (sinusoid regression, standard Omniglot, and *mini*ImageNet) for completeness in Appendix E.

### 5.2 Module discovery

To determine whether shrinkage discovers task-specific modules, we examine the learned prior strengths of each module and then adapt individual (or a subset of the) modules to assess the effect on performance due to adapting the chosen modules. In these experiments, we treat each layer as a module but other choices of modules are straightforward.

**Image classification.** Fig. 2 shows our module discovery results on large-data augmented Omniglot for $\sigma$-iMAML and $\sigma$-Reptile using the standard network (4 conv layers, 4 batch-norm layers, and a linear output layer). In each case, the learned $\sigma^2$ (Fig. 2(a,b)) of the output layer is considerably larger than the others. Fig. 2(c,d) show that the model achieves high accuracy when adapting only this shrinkage-identified module, giving comparable performance to that achieved by adapting all layers according to $\sigma^2$. This corroborates the conventional belief that the output layers of image classification networks are more task-specific while the input layers are more general, and bolsters other meta-learning studies [17, 25] that propose to adapt only the output layer.

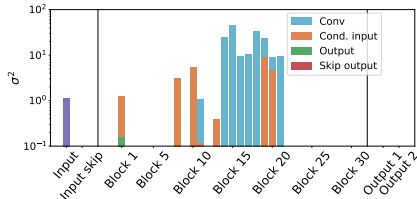

Figure 3: Learned $\sigma^2$ of WaveNet modules. Every block contains four layers (See Appendix E.2 for details).

| | |
|---|---|
| $\sigma$-iMAML | $73.6 \pm 1.3\%$ |
| iMAML | $72.8 \pm 1.2\%$ |
| $\sigma$-Reptile | $78.9 \pm 1.2\%$ |
| Reptile | $77.8 \pm 1.1\%$ |

Table 2: Average test accuracy and $95\%$ confidence intervals for 10 runs on large-data augmented Omniglot.

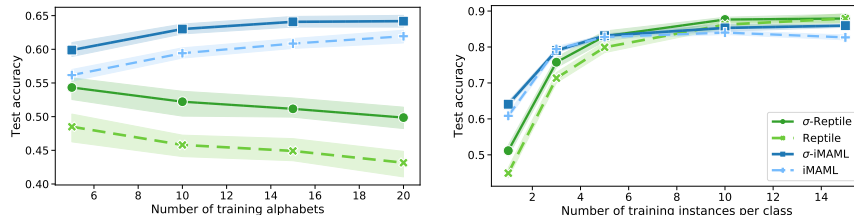

Figure 4: Mean test accuracy and $95\%$ confidence intervals for 10 runs on small-data aug. Omniglot as a function of the number of alphabets (left, 1 image per character) and instances (right, 15 alphabets).

However, the full story is not so clear cut. Our module discovery results (Appendix E) on standard few-shot short-adaptation image classification show that in those domains adapting the *penultimate* layer is best, which matches an observation in Arnold et al. [19]. Further, on sinusoid regression, adapting the *first* layer performed best. Thus, there is no single best modular structure across domains.

**Text-to-speech.** Fig. 3 shows the learned $\sigma^2$ for each layer of our TTS WaveNet model, which consists of 4 layers per residual block and 123 layers in total (Appendix E.2 shows the full architecture). Most $\sigma^2$ values are too small to be visible. The dilated conv layers between blocks 10 and 21 have the largest $\sigma^2$ values and thus require the most adaptability, suggesting that these blocks model the most speaker-specific features. These layers have a receptive field of 43–85 ms, which matches our intuition about the domain because earlier blocks learn to model high-frequency sinusoid-like waveforms and later blocks model slow-changing prosody. WaveNet inputs include the fundamental frequency (f0), which controls the change of pitch, and a sequence of linguistic features that provides the prosody. Both the earlier and later residual blocks can learn to be speaker-invariant given these inputs. Therefore, it is this middle range of temporal variability that contains the information about speaker identity. We select the 12 layers with $\sigma^2$ values above 3.0 for adaptation below. This requires adding only 16% of the network parameters for each new voice. Note that this domain exhibits yet another type of modular structure from those above.

## 5.3 Predictive Performance

**Image classification accuracy.** For each algorithm, we perform extensive hyperparameter tuning on validation data. Details are provided in Appendix E. Table 2 shows the test accuracy for augmented Omniglot in the large-data regime. Both shrinkage variants obtain modest accuracy improvements over their non-modular counterparts. We expect only this small improvement over the non-shrinkage variants, however, as the heavy data augmentation in this domain reduces overfitting.

We now reduce the amount of augmentation and data to make the domain more challenging. Fig. 4 shows our results in this small-data regime. Both shrinkage variants significantly improve over their non-shrinkage counterparts when there are few training instances per alphabet. This gap grows as the number of training instances decreases, demonstrating that shrinkage helps prevent overfitting. Interestingly, the Reptile variants begin to outperform the iMAML variants as the number of training instances increases, despite the extra validation data used by the iMAML variants. Results for all combinations of alphabets and instances are shown in the appendix.

**Text-to-speech sample quality.** The state-of-the art approaches for this domain [1] are to finetune either the entire model (aka. SEA-All) or just the speaker embedding (SEA-Emb). We compare these two methods to meta-training with Reptile and $\sigma$-Reptile. We also tried to run $\sigma$-MAML and $\sigma$-iMAML but $\sigma$-MAML ran out of memory with one adaptation step and $\sigma$-iMAML trained too slowly.

We evaluate the generated sample quality using two voice synthesis metrics: (1) the voice similarity between a sample and real speaker utterances using a speaker verification model [47, 1], and (2) the sample naturalness measured by the mean opinion score (MOS) from human raters. Fig. 5 shows the distribution of sample similarities for each method, along with an upper (lower) bound computed from real utterances between the same (different) speakers. Sample naturalness for each method is shown in Table 3, along with an upper bound created by training the same model on 40 hours of data.

$\sigma$-Reptile and Reptile clearly outperform SEA-All and SEA-Emb. $\sigma$-Reptile has comparable median similarity with Reptile, and its sample naturalness surpasses Reptile with both 8 minutes of speech data, and 4 minutes, which is less than used in meta-training. Overall, the $\sigma$-Reptile samples have the highest quality despite adapting only 12 of the 123 modules. SEA-All and Reptile, which adapt all modules, overfit quickly and underperform, despite adaptation being early-stopped. Conversely, SEA-Emb underfits and does not improve with more data because it only tunes the speaker embedding.

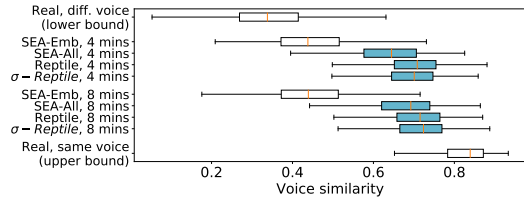

|  | 4 mins | 8 mins |
|---|---|---|
| SEA-Emb | $1.51 \pm 0.05$ | $1.57 \pm 0.05$ |
| SEA-All | $1.41 \pm 0.04$ | $1.73 \pm 0.06$ |
| Reptile | $\mathbf{1.93} \pm 0.05$ | $2.09 \pm 0.06$ |
| $\sigma$-Reptile | $\mathbf{1.98} \pm 0.06$ | $\mathbf{2.28} \pm 0.06$ |
| Trained with 40 hours of data (upper bound) | $2.59 \pm 0.07$ | |

Figure 5: Box-plot of voice similarity measurements from utterances (higher is better).

Table 3: Mean opinion score of sample naturalness. Scores range from 1–5 (higher is better).

## 5.4 Discussion

We thus answer all three experimental questions in the affirmative. In both image classification and text-to-speech, the learned shrinkage priors correspond to meaningful and interesting task-specific modules. These modules differ between domains, however, indicating that they should be learned from data. Studying these learned modules allows us to discover new or existing knowledge about the behavior of different parts of the network, while adapting only the task-specific modules provides the same performance as adapting all layers. Finally, learning and using our shrinkage prior helps prevent overfitting and improves performance in low-data, long-adaptation regimes.

## 6 Conclusions

This work proposes a hierarchical Bayesian model for meta-learning that places a shrinkage prior on each module to allow learning the extent to which each module should adapt, without a limit on the adaptation horizon. Our formulation includes MAML, Reptile, and iMAML as special cases, empirically discovers a small set of task-specific modules in various domains, and shows promising improvement in a practical TTS application with low data and long task adaptation. As a general modular meta-learning framework, it allows many interesting extensions, including incorporating alternative Bayesian inference algorithms, modular structure learning, and learn-to-optimize methods.

## Broader Impact

This paper presents a general meta-learning technique to automatically identify task-specific modules in a model for few-shot machine learning problems. It reduces the need for domain experts to hand-design task-specific architectures, and thus further democratizes machine learning, which we hope will have a positive societal impact. In particular, general practitioners who can not afford to collect a large amount of labeled data will be able to take advantage of a pre-trained generic meta-model and adapt its task-specific components to a new task based on limited data. One example application might be to adapt a multilingual text-to-speech model to a low-resource language or the dialect of a minority ethnic group.

As a data-driven method, like other machine learning techniques, the task-independent and task-specific modules discovered by our method are based on the distribution of tasks in the meta-training phase. Adaptation may not generalize to a task with characteristics that fundamentally differ from those of the training distribution. Applying our method to a new task without examining the task

similarity runs the risk of transferring induced bias from meta-training to the out-of-distribution task. For example, a meta image classification model trained only on vehicles is unlikely to be able to be finetuned to accurately identify a pedestrian based on the adaptable modules discovered during meta-training. To mitigate this problem, we suggest ML practitioners first understand whether the characteristics of the new task match those of the training task distribution before applying our method.

## Acknowledgments and Disclosure of Funding

The authors thank Alexander Novikov for his code for computing Hessian-vector products; Yi Yang for helpful discussions on the meta-learning dataset design; Sander Dieleman, Norman Casagrande and Chenjie Gu for their advice on the TTS model; and Sarah Henderson and Claudia Pope for their help.

All authors are employees of DeepMind, which was the sole source of funding for this work. None of the authors have competing interests.

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
