[Supplementary Material]

# Appendix

This appendix contains the supplementary material for the main text. In Appendix A, we first prove Lemma 1 and then provide a detailed analysis of the introduced estimation approaches on a simple example. In Appendix B, we provide details of the derivation of the implicit gradients in Eqs. (7) and (10), show the equivalence of $\boldsymbol{\sigma}$-iMAML and iMAML when $\sigma_m^2$ is shared across all modules and fixed, and provide more discussion on the objective introduced in Section 4.2. In Appendix C, we demonstrate the different behavior of approaches that meta-learn the learning rate per module versus our approach that meta-learns the shrinkage prior per module, on two synthetic problems. In Appendix D, we provide additional details about our implementation of the iMAML baseline and our shrinkage algorithms. Finally, in Appendix E, we explain the experiments in more detail, provide information about the setup and hyperparameters, and present additional results.

## A  Analysis

In this section, we first prove Lemma 1, which shows in general that it is not feasible to estimate all the parameters $(\boldsymbol{\sigma}^2, \boldsymbol{\phi}, \boldsymbol{\theta}_{1:T})$ jointly by optimizing the joint log-density. We then provide detailed analysis of the properties of the two estimation approaches introduced in Section 4 for a simple hierarchical normal distribution example. Finally, we discuss the pathological behavior when estimating $\boldsymbol{\sigma}^2$ from the joint log-density in the same example.

### A.1  Maximization of $\log p(\boldsymbol{\theta}_{1:T}, \mathcal{D} | \boldsymbol{\sigma}^2, \boldsymbol{\phi})$

Lemma 1 states that the function $f : (\boldsymbol{\sigma}^2, \boldsymbol{\phi}, \boldsymbol{\theta}_{1:T}) \mapsto \log p(\boldsymbol{\theta}_{1:T}, \mathcal{D} | \boldsymbol{\sigma}^2, \boldsymbol{\phi})$ diverges to $+\infty$ as $\boldsymbol{\sigma} \to 0^+$ when $\theta_{t,m} = \phi_m$ for all $t \in \{1, ..., T\}, m \in \{1, ..., M\}$. We first establish here this result.

*Proof of Lemma 1.* For simplicity, we prove the result for $M = 1$ and $N_t = N$. The extension to the general case is straightforward. We have

$$\log p(\theta_{1:T}, \mathbf{x}_{1:T} | \phi, \sigma^2) = -\frac{T}{2} \log \sigma^2 - \frac{1}{2} \sum_{t=1}^{T} \frac{(\theta_t - \phi)^2}{\sigma^2} + \sum_{t=1}^{T} \sum_{i=1}^{N} \log p(x_{t,n} | \theta_t).$$

From this expression, it is clear that when $\theta_t = \phi$ for all $t$, then $\log p(\theta_{1:T}, \mathbf{x}_{1:T} | \phi, \sigma^2) \to +\infty$ as $\sigma^2 \to 0^+$.  □

This shows that the global maximum of $\log p(\theta_{1:T}, \mathbf{x}_{1:T} | \phi, \sigma^2)$ does not exist in general, and thus we should not use this method for estimating $\sigma^2$.

This negative result illustrates the need for alternative estimators. In the following sections, we analyze the asymptotic behavior of the estimates of $\boldsymbol{\phi}$ and $\boldsymbol{\sigma}^2$ proposed in Section 4 as the number of training tasks $T \to \infty$ in a simple example. In Appendix A.2.3, we analyze the behavior of Lemma 1 when optimizing $\log p(\theta_{1:T}, \mathbf{x}_{1:T} | \phi, \sigma^2)$ w.r.t. $\boldsymbol{\sigma}^2$ with a local optimizer.

### A.2  Analysis of estimates in a simple example

To illustrate the asymptotic behavior of different learning strategies as $T \to \infty$, we consider a simple model with $M = 1$ module, $D_t = D = 1$ for all tasks and normally distributed observations. We use non-bolded symbols to denote that all variables are scalar in this section.

**Example 1** (Univariate normal)**.**

$$M = 1, D_t = 1, N_t = N, \forall t = 1, \ldots, T,$$
$$x_{t,n} \sim \mathcal{N}(x_{t,n} | \theta_t, 1), \forall t = 1, \ldots, T, n = 1, \ldots, N.$$

It follows that

$$p(\theta_{1:T}, \mathbf{x}_{1:T} | \phi, \sigma^2) = \prod_{t=1}^{T} \left( \mathcal{N}(\theta_t | \phi, \sigma^2) \prod_{n=1}^{N} \mathcal{N}(x_{t,n} | \theta_t, 1) \right), \tag{11}$$

and we use the notation for the negative joint log-density up to a constant

$$\ell_{\text{joint}}(\theta_{1:T}, \phi, \sigma^2) = -\log p(\theta_{1:T}, \mathbf{x}_{1:T}|\phi, \sigma^2)$$

$$= \frac{T}{2}\log\sigma^2 + \frac{1}{2}\sum_{t=1}^{T}\frac{(\theta_t - \phi)^2}{\sigma^2} + \frac{1}{2}\sum_{t=1}^{T}\sum_{n=1}^{N}(x_{t,n} - \theta_t)^2$$

$$= \sum_{t=1}^{T}\ell_t^{\text{train}}(\theta_t, \phi, \sigma^2). \tag{12}$$

We also assume there exists a set of independently sampled validation data $\mathbf{y}_{1:T}$ where $y_{t,n} \sim \mathcal{N}(y|\theta_t, 1)$ for $t \in \{1, \ldots, T\}, n \in \{1, \ldots, K\}$. The corresponding negative log-likelihood given point estimates $\hat{\theta}_{1:T}$ is given up to a constant by

$$\hat{\ell}_{\text{PLL}} = \sum_{t=1}^{T}\ell_t^{\text{val}}(\hat{\theta}_t) = \frac{1}{2}\sum_{t=1}^{T}\sum_{k=1}^{K}(y_{t,k} - \hat{\theta}_t)^2. \tag{13}$$

We denote by $\phi_{\text{True}}$ and $\sigma_{\text{True}}$ the true value of $\phi$ and $\sigma$ for the data generating process described above.

### A.2.1 Estimating $\phi$ and $\phi$ with predictive log-likelihood

We first show that when we estimate $\theta_t$ with MAP on $\ell_t^{\text{train}}$ and estimate $\phi$ and $\sigma^2$ with the predictive log-likelihood as described in Section 4.1, the estimates $\hat{\phi}$ and $\hat{\sigma}^2$ are consistent.

**Proposition 1.** *Let* $\hat{\theta}_t(\phi, \sigma^2) = \arg\min_{\theta_t}\ell_t^{\text{train}}(\theta_t, \phi, \sigma^2)$ *and define* $(\hat{\phi}, \hat{\sigma}^2) = \arg_{\phi,\sigma^2}\{\nabla_{(\phi,\sigma^2)}\hat{\ell}_{\text{PLL}}(\hat{\theta}_{1:T}) = 0\}$ *then, as* $T \to \infty$, *we have*

$$\hat{\phi}(\hat{\sigma}^2) \to \phi_{\text{True}}, \quad \hat{\sigma}^2 \to \sigma_{\text{True}}^2,$$

*in probability.*

*Proof.* We denote the sample average

$$\bar{x}_t := \frac{1}{N}\sum_{n,t}x_{n,t}, \quad \bar{y}_t := \frac{1}{K}\sum_{k,t}y_{k,t}, \tag{14}$$

and the average over all tasks $\bar{x} = \frac{1}{T}\sum_{t=1}^{T}\bar{x}_t, \bar{y} = \frac{1}{T}\sum_{t=1}^{T}\bar{y}_t$.

The equation $\nabla_{\theta_t}\ell_t^{\text{train}} = 0$ gives

$$\hat{\theta}_t(\phi, \sigma^2) = \frac{\sum_{n=1}^{N}x_{t,n} + \phi/\sigma^2}{N + 1/\sigma^2} = \frac{\bar{x}_t + \phi/N\sigma^2}{1 + 1/N\sigma^2}. \tag{15}$$

By plugging Eq. (15) in Eq. (7), it follows that

$$\nabla_{\sigma^2}\hat{\ell}_{\text{PLL}} = -\sum_t\nabla_{\theta_t}\ell_t^{\text{val}}(\theta_t)\mathbf{H}_{\theta_t\theta_t}^{-1}\mathbf{H}_{\theta_t\sigma^2}$$

$$= \frac{K}{N\sigma^4\left(1 + \frac{1}{N\sigma^2}\right)}\sum_t(\theta_t - \bar{y}_t)(\theta_t - \phi)$$

$$= \frac{K}{N\sigma^4(1 + \frac{1}{N\sigma^2})^3}.$$

$$\sum_t\left(\bar{x}_t - \bar{y}_t + \frac{1}{N\sigma^2}(\phi - \bar{y}_t)\right)(\bar{x}_t - \phi). \tag{16}$$

We then solve $\nabla_{\sigma^2}\hat{\ell}_{\text{PLL}} = 0$ as a function of $\phi$,

$$\hat{\sigma}^2(\phi) = \frac{\frac{1}{T}\sum_t(\bar{x}_t - \phi)(\bar{y}_t - \phi)}{\frac{N}{T}\sum_t(\bar{x}_t - \bar{y}_t)(\bar{x}_t - \phi)}. \tag{17}$$

Similarly, we solve $\nabla_\phi \hat{\ell}_{\text{PLL}}(\hat{\theta}_{1:T}(\phi, \sigma^2) = 0$ as

$$\nabla_\phi \hat{\ell}_{\text{PLL}} = -\sum_t \nabla_{\theta_t} \ell_t^{\text{val}}(\theta_t) \mathbf{H}_{\theta_t \theta_t}^{-1} \mathbf{H}_{\theta_t \phi}$$

$$= -\frac{K}{1 + N\sigma^2} \sum_{t=1}^T \left( \bar{y}_t - \frac{\bar{x}_t + \phi/N\sigma^2}{1 + 1/N\sigma^2} \right) = 0,$$

(18)

this yields

$$\hat{\phi}(\sigma^2) = \bar{y} + N\sigma^2(\bar{y} - \bar{x}).$$

(19)

By combining Eq. (17) to Eq. (19), we obtain

$$\hat{\phi} = \frac{\overline{x(x-y)}\bar{y} + \overline{xy}(\bar{y} - \bar{x})}{\bar{x}(\bar{y} - \bar{x}) + \overline{x(x-y)}}$$

(20)

where, for a function $f$ of variable $x, y$, we define $\overline{f(x,y)} := \frac{1}{T} \sum_t f(\bar{x}_t, \bar{y}_t)$.

Following the data generating process, the joint distribution of $\bar{x}_t$ and $\bar{y}_t$ with $\theta_t$ integrated out is jointly normal and satisfies

$$[\bar{x}_t, \bar{y}_t]^T \sim \mathcal{N} \left( \phi_{\text{True}} \mathbf{1}_2, \begin{bmatrix} \sigma_{\text{True}}^2 + \frac{1}{N} & \sigma_{\text{True}}^2 \\ \sigma_{\text{True}}^2 & \sigma_{\text{True}}^2 + \frac{1}{K} \end{bmatrix} \right).$$

(21)

As $T \to \infty$, it follows from the law of large numbers and Slutsky's lemma that $\hat{\phi} \to \phi_{\text{True}}$ in probability. Consequently, it also follows from (17) that $\hat{\sigma}^2 \to \sigma_{\text{True}}^2$ in probability. $\square$

### A.2.2 Estimating $\phi$ with MAP and $\sigma^2$ with predictive log-likelihood

Alternatively, we can follow the approach described in Section 4.2 to estimate both $\theta_{1:T}$ and $\phi$ with MAP on $\ell_{\text{joint}}$ (Eq. (12)), i.e.,

$$(\hat{\phi}(\sigma^2), \hat{\theta}_{1:T}(\sigma^2)) = \arg \max_{(\phi, \theta_{1:t})} \ell_{\text{joint}}(\theta_{1:T}, \phi, \sigma^2),$$

(22)

and estimate $\sigma^2$ with the predictive log-likelihood by finding the root of the approximate implicit gradient in Eq. (10).

We first show that given any fixed value of $\sigma^2$, the MAP estimate of $\phi$ is consistent.

**Proposition 2** (Consistency of MAP estimate of $\phi$). *For any fixed $\sigma > 0$, $\hat{\phi}(\sigma^2) = \bar{x}$, so $\hat{\phi}(\sigma^2) \to \phi_{\text{True}}$ in probability as $T \to \infty$.*

*Proof of Proposition 2.* The equation $\nabla_\phi \ell_{\text{joint}} = 0$ for $\ell_{\text{joint}}$ defined in Eq. (12) gives

$$\hat{\phi}(\sigma^2) = \frac{1}{T} \sum_{t=1}^T \hat{\theta}_t(\sigma^2).$$

(23)

By summing Eq. (15) over $t = 1, ..., T$ and using Eq. (23), we obtain

$$\hat{\phi}(\sigma^2) = \bar{x}, \quad \hat{\theta}_t(\sigma^2) = \frac{\bar{x}_t + \frac{\bar{x}}{N\sigma^2}}{1 + \frac{1}{N\sigma^2}}.$$

(24)

The distribution of $\bar{x}$ follows directly from Eq. (21). Therefore, $\hat{\phi}(\sigma^2)$ is unbiased for any $T$ and it is additionally consistent by Chebyshev's inequality as $T \to \infty$. $\square$

Now we show that estimating $\sigma^2$ by finding the roots of the implicit gradient in Eq. (10) is also consistent.

**Proposition 3.** *Let* $(\hat{\phi}(\sigma^2), \hat{\theta}_{1:T}(\sigma^2)) = \arg\max_{(\phi, \theta_{1:T})} \ell_{\text{joint}}(\theta_{1:T}, \phi, \sigma^2)$ *and define* $\hat{\sigma}^2 = \arg_{\sigma^2}\{\nabla_{\sigma^2}\hat{\ell}_{\text{PLL}}(\hat{\theta}_{1:T}) = 0\}$ *where the gradient is defined as in Eq.* (10), *then, as* $T \to \infty$, *we have*

$$\hat{\phi}(\hat{\sigma}^2) \to \phi_{\text{True}}, \quad \hat{\sigma}^2 \to \sigma^2_{\text{True}},$$

*in probability.*

*Proof.* From Eq. (10), we follow the same derivation as Eq. (17) and get the root of the gradient for $\sigma^2$ as a function of $\hat{\phi}$:

$$\hat{\sigma}^2(\hat{\phi}) = \frac{\frac{1}{T}\sum_t(\bar{x}_t - \hat{\phi})(\bar{y}_t - \hat{\phi})}{\frac{N}{T}\sum_t(\bar{x}_t - \bar{y}_t)(\bar{x}_t - \hat{\phi})}. \tag{25}$$

By plugging Eq. (24), it follows from the joint distribution of $(\bar{x}_t, \bar{y}_t)$ (Eq. 21), the law of large numbers and Slutsky's lemma that $\hat{\sigma}^2 \to \sigma^2_{\text{True}}$ in probability as $T \to \infty$. $\square$

### A.2.3 MAP estimate of $\sigma^2$

From Lemma 1, we know that maximizing $\ell_{\text{joint}}$ (Eq. 12) w.r.t. $(\theta_t, \phi, \sigma^2)$ is bound to fail. Here we show the specific $\sigma^2$ estimate one would obtain by following gradient descent on $\ell_{\text{joint}}$ in the running example.

Let $S$ denote the sample variance of $\bar{x}_t$ across tasks, that is

$$S = \frac{\sum_{t=1}^T(\bar{x}_t - \bar{x})^2}{T}. \tag{26}$$

We can easily establish from Eq. (21) that

$$S \sim \frac{\sigma^2_{\text{True}} + \frac{1}{N}}{T}\chi^2_{T-1}, \tag{27}$$

where $\chi^2_{T-1}$ is the standard Chi-squared random variable with $T-1$ degrees of freedom. Although $\hat{\phi}(\sigma^2)$ is consistent whenever $\sigma^2 > 0$, the following proposition shows that maximizing $\ell_{\text{joint}}(\hat{\theta}_{1:T}(\sigma^2), \hat{\phi}(\sigma^2), \sigma^2)$ w.r.t. $\sigma^2$ remains problematic.

**Proposition 4** (Estimation of $\sigma$ by gradient descent). *Minimizing the function* $\sigma^2 \mapsto \ell_{\text{joint}}(\hat{\theta}_{1:T}(\sigma^2), \hat{\phi}(\sigma^2), \sigma^2)$ *by gradient descent will diverge at* $\sigma \to 0^+$ *if either of the following two conditions is satisfied*

1. $S < \frac{4}{N}$,

2. $\sigma^2$ *is initialized in* $\left(0, \frac{1}{2}\left(S - \frac{2}{N} - \sqrt{S(S - \frac{4}{N})}\right)\right)$.

*Otherwise, it converges to a local minimum* $\hat{\sigma}^2 = \frac{1}{2}\left(S - \frac{2}{N} + \sqrt{S(S - \frac{4}{N})}\right)$.

**Corollary 1.** *As the number of training tasks* $T \to \infty$, *condition 1 is equivalent to*

$$\sigma^2_{\text{True}} < \frac{3}{N}$$

*while the upper endpoint of the interval in condition 2 becomes*

$$\frac{1}{2}\left(\sigma^2_{\text{True}} - \frac{1}{N} - \sqrt{(\sigma^2_{\text{True}} + \frac{1}{N})(\sigma^2_{\text{True}} - \frac{3}{N})}\right),$$

*beyond which the* $\sigma^2$ *estimate converges to*

$$\hat{\sigma}^2 = \frac{1}{2}\left(\sigma^2_{\text{True}} - \frac{1}{N} + \sqrt{(\sigma^2_{\text{True}} + \frac{1}{N})(\sigma^2_{\text{True}} - \frac{3}{N})}\right). \tag{28}$$

*Proofs of Proposition 4 and Corollary 1.* It follows from Eq. (24) that

$$\hat{\theta}_t(\sigma) - \hat{\phi}(\sigma) = \frac{\bar{x}_t - \bar{x}}{1 + \frac{1}{N\sigma^2}}. \tag{29}$$

By solving $\nabla_{\sigma^2} \ell_{\text{joint}} = 0$, we obtain

$$\hat{\sigma}^2 = \frac{\sum_{t=1}^T (\theta_t - \phi)^2}{T}. \tag{30}$$

Plugging Eq. (29) into this expression yields

$$\hat{\sigma}^2 = \frac{\sum_{t=1}^T (\bar{x}_t - \bar{x})^2}{T(1 + \frac{1}{N\hat{\sigma}^2})^2} = \frac{S}{(1 + \frac{1}{N\hat{\sigma}^2})^2}. \tag{31}$$

Hence, by rearranging this expression, we obtain the following quadratic equation for $\hat{\sigma}^2$

$$\hat{\sigma}^4 + (2/N - S)\hat{\sigma}^2 + 1/N^2 = 0. \tag{32}$$

Positive roots of Eq. (32) exist if and only if

$$S \geq \frac{4}{N}. \tag{33}$$

When condition (33) does not hold, no stationary point exists and gradient descent from any initialization will diverge toward $\hat{\sigma}^2 \to 0^+$ as it can be checked that $\nabla_{\sigma^2} \ell_{\text{joint}} < 0$. Figs. 6a and 6c illustrate $\ell_{\text{joint}}$ and $\nabla_{\sigma^2} \ell_{\text{joint}}$ as a function of $\sigma^2$ in this case.

When the condition above is satisfied, there exist two (or one when $S = 4/N$, an event of zero probability) roots at:

$$\sigma_{\text{root}}^2 = \frac{1}{2}\left(S - \frac{2}{N} \pm \sqrt{S(S - \frac{4}{N})}\right). \tag{34}$$

By checking the sign of the gradient $\nabla_{\sigma^2} \ell_{\text{joint}}$ and plugging it in to Eq. (29), we find that the left root is a local maximum and the right root is a local minimum. So if one follows gradient descent to estimate $\sigma^2$, it will converge towards $0^+$ when $\sigma^2$ is initialized below the left root and to the second root otherwise. Figs. 6b and 6d illustrate the function of $\ell_{\text{joint}}$ and its gradient as a function of $\sigma^2$ when $\phi$ and $\theta_t$ are at their stationary point and condition 1 is satisfied.

To prove the corollary, we note that it follows from Eq. (21) that $S \to \sigma_{\text{True}}^2 + \frac{1}{N}$ as $T \to \infty$. Hence the condition Eq. (33) approaches

$$\sigma_{\text{True}}^2 \geq \frac{3}{N}. \tag{35}$$

Similarly, when $T \to \infty$, Eq. (35) becomes

$$\sigma_{\text{root}}^2 = \frac{1}{2}\left(\sigma_{\text{True}}^2 - \frac{1}{N} \pm \sqrt{(\sigma_{\text{True}}^2 + \frac{1}{N})(\sigma_{\text{True}}^2 - \frac{3}{N})}\right). \tag{36}$$

$\square$

# B    Derivation of the approximate gradient of predictive log-likelihood in Section 4

## B.1    Implicit gradient of $\sigma$-iMAML in Eq. (7)

**Lemma 2.** *(Implicit differentiation) Let $\hat{\mathbf{y}}(\mathbf{x})$ be the stationary point of function $f(\mathbf{x}, \mathbf{y})$, i.e. $\nabla_{\mathbf{y}} f(\mathbf{x}, \mathbf{y})|_{\mathbf{y}=\hat{\mathbf{y}}(\mathbf{x})} = 0 \; \forall \mathbf{x}$, then the gradient of $\hat{\mathbf{y}}$ w.r.t. $\mathbf{x}$ can be computed as*

$$\nabla_{\mathbf{x}} \hat{\mathbf{y}}(\mathbf{x}) = -(\nabla_{\mathbf{yy}}^2 f)^{-1} \nabla_{\mathbf{yx}}^2 f. \tag{37}$$

(a) $\ell_{\mathrm{joint}}(\sigma^2)$ when stationary points do not exist.

(b) $\ell_{\mathrm{joint}}(\sigma^2)$ when stationary points exist.

(c) $\nabla_{\sigma^2}\ell_{\mathrm{joint}}$ when stationary points do not exist.

(d) $\nabla_{\sigma^2}\ell_{\mathrm{joint}}$ when stationary points exist.

Figure 6: Example of $\ell_{\mathrm{joint}}(\sigma^2)$ up to a constant, and its gradient w.r.t. $\sigma^2$. Orange dots denote stationary points.

By applying the chain rule, the gradient of the approximate predictive log-likelihood $\ell_t^{\mathrm{val}}(\hat{\boldsymbol{\theta}}_t)$ (from Eq. (5)) w.r.t. the meta variables $\boldsymbol{\Phi} = (\sigma^2, \boldsymbol{\phi})$ is given by

$$\nabla_{\boldsymbol{\Phi}}\ell_t^{\mathrm{val}}(\hat{\boldsymbol{\theta}}_t(\boldsymbol{\Phi})) = \nabla_{\hat{\boldsymbol{\theta}}_t}\ell_t^{\mathrm{val}}(\hat{\boldsymbol{\theta}}_t)\nabla_{\boldsymbol{\Phi}}\hat{\boldsymbol{\theta}}_t(\boldsymbol{\Phi}). \tag{38}$$

Applying Lemma 2 to the joint log-density on the training subset in Eq. (4), $\ell_t^{\mathrm{train}}(\boldsymbol{\theta}_t, \boldsymbol{\Phi})$. We have

$$\nabla_{\boldsymbol{\Phi}}\hat{\boldsymbol{\theta}}_t(\boldsymbol{\Phi}) = -\left(\nabla^2_{\boldsymbol{\theta}_t\boldsymbol{\theta}_t}\ell_t^{\mathrm{train}}\right)^{-1}\nabla^2_{\boldsymbol{\theta}_t\boldsymbol{\Phi}}\ell_t^{\mathrm{train}}. \tag{39}$$

Plug the equation above to Eq. (38) and we obtain the implicit gradient of $\ell_t^{\mathrm{val}}$ in Eq. (7).

## B.2  Equivalence between $\sigma$-iMAML and iMAML when $\sigma_m^2$ is constant

When all modules share a constant variance, $\sigma_m^2 \equiv \sigma^2$, we expand the log-prior term for task parameters $\boldsymbol{\theta}_t$ in $\ell_t^{\mathrm{train}}$ (4) and plug in the normal prior assumption as follows,

$$\begin{aligned}
\log p(\boldsymbol{\theta}_t|\sigma^2, \boldsymbol{\phi}) &= \sum_{m=1}^{M}\left(-\frac{D_m}{2}\log(2\pi\sigma_m^2) - \frac{\|\boldsymbol{\theta}_{mt} - \boldsymbol{\phi}_m\|^2}{2\sigma_m^2}\right)\\
&= -\frac{D}{2}\log(2\pi\sigma^2) - \frac{\|\boldsymbol{\theta}_t - \boldsymbol{\phi}\|^2}{2\sigma^2}.
\end{aligned} \tag{40}$$

By plugging the equation above to Eq. (7), we obtain the update for $\boldsymbol{\phi}$ as

$$\begin{aligned}
\Delta_t^{\boldsymbol{\sigma}\text{-iMAML}} &= \nabla_{\boldsymbol{\theta}_t}\ell_t^{\mathrm{val}}(\boldsymbol{\theta}_t)\left(\frac{1}{\sigma^2}\mathbf{I} - \nabla^2_{\boldsymbol{\theta}_t\boldsymbol{\theta}_t}\log p(\mathcal{D}_t^{\mathrm{train}}|\boldsymbol{\theta}_t)\right)^{-1}\frac{1}{\sigma^2}\mathbf{I}\\
&= \nabla_{\boldsymbol{\theta}_t}\ell_t^{\mathrm{val}}(\boldsymbol{\theta}_t)\left(\mathbf{I} - \sigma^2\nabla^2_{\boldsymbol{\theta}_t\boldsymbol{\theta}_t}\log p(\mathcal{D}_t^{\mathrm{train}}|\boldsymbol{\theta}_t)\right)^{-1}.
\end{aligned} \tag{41}$$

This is equivalent to the update of iMAML by defining the regularization scale $\lambda = 1/\sigma^2$ and plugging in the definition of $\ell_t^{\mathrm{train}} := -\log p(\mathcal{D}_t^{\mathrm{train}}|\boldsymbol{\theta}_t)$ in Section 2.

**B.3 Discussion on the alternative procedure for Bayesian parameter learning (Section 4.2)**

By plugging the MAP of $\phi$ (Eq. (9)) into Eq. (8) and scaling by $1/T$, we derive the approximate predictive log-likelihood as

$$\hat{\ell}_{\text{PLL}}(\boldsymbol{\sigma}^2) = \frac{1}{T}\sum_{t=1}^{T} \ell_t^{\text{val}}(\hat{\boldsymbol{\theta}}_t(\boldsymbol{\sigma}^2)). \tag{42}$$

It is a sensible strategy to estimate both the task parameters $\boldsymbol{\theta}_{1:T}$ and the prior center $\phi$ with MAP on the training joint log-density and estimate the prior variance $\boldsymbol{\sigma}^2$ on the predictive log-likelihood. If $\hat{\phi}(\boldsymbol{\sigma}^2) \to \bar{\phi}(\boldsymbol{\sigma}^2)$ as $T \to \infty$ we can think of both $\ell_{\text{PLL}}(\boldsymbol{\sigma}^2)$ and $\hat{\ell}_{\text{PLL}}(\boldsymbol{\sigma}^2)$ as approximations to

$$\tilde{\ell}_{\text{PLL}}(\boldsymbol{\sigma}^2) = -\frac{1}{T}\sum_{t=1}^{T}\log p(\mathcal{D}_t^{\text{val}}|\mathcal{D}_t^{\text{train}}, \bar{\phi}(\boldsymbol{\sigma}^2), \boldsymbol{\sigma}^2), \tag{43}$$

which, for $(\mathcal{D}_t^{\text{train}}, \mathcal{D}_t^{\text{val}}) \overset{\text{i.i.d.}}{\sim} \nu$, converges by the law of large numbers as $T \to \infty$ towards

$$\tilde{\ell}_{\text{PLL}}(\boldsymbol{\sigma}^2) \to -\mathbb{E}_{\nu(\mathcal{D}_t^{\text{train}}, \mathcal{D}_t^{\text{val}})}[\log p(\mathcal{D}_t^{\text{val}}|\mathcal{D}_t^{\text{train}}, \bar{\phi}(\boldsymbol{\sigma}^2), \boldsymbol{\sigma}^2)]. \tag{44}$$

Similarly to Eq. (3), it can be shown that minimizing the r.h.s. of Eq. (42) is equivalent to minimizing the average KL

$$\mathbb{E}_{\nu(\mathcal{D}_t^{\text{train}})}[\text{KL}(\nu(\mathcal{D}_t^{\text{val}}|\mathcal{D}_t^{\text{train}})||p\left(\mathcal{D}_t^{\text{val}}|\mathcal{D}_t^{\text{train}}, \bar{\phi}(\boldsymbol{\sigma}^2), \boldsymbol{\sigma}^2)\right)]. \tag{45}$$

**B.4 Meta update of $\sigma$-Reptile in Eq. (10)**

The meta update for $\phi$ can then be obtained by differentiating (9) with respect to $\phi$.

To derive the gradient of Eq. (42) with respect to $\boldsymbol{\sigma}$, notice that when $\phi$ is estimated as the MAP on the training subsets of all tasks, it becomes a function of $\boldsymbol{\sigma}^2$. Denote by $\ell^{\text{joint}}(\boldsymbol{\Theta}, \boldsymbol{\sigma}^2)$ the objective in Eq. (9) where $\boldsymbol{\Theta} = (\boldsymbol{\theta}_{1:T}, \phi)$ is the union of all task parameters $\boldsymbol{\theta}_t$ and the prior central $\phi$. It requires us to apply the implicit function theorem to $\ell^{\text{joint}}(\boldsymbol{\Theta}, \boldsymbol{\sigma}^2)$ in order to compute the gradient of the approximate predictive log-likelihood w.r.t. $\boldsymbol{\sigma}^2$. However, the Hessian matrix $\nabla^2_{\boldsymbol{\Theta},\boldsymbol{\Theta}}\ell^{\text{joint}}$ has a size of $D(T+1) \times D(T+1)$ where $D$ is the size of a model parameter $\boldsymbol{\theta}_t$, which becomes too expensive to compute when $T$ is large.

Instead, we take an approximation and ignore the dependence of $\hat{\phi}$ on $\boldsymbol{\sigma}^2$. Then $\phi$ becomes a constant when computing $\nabla_{\boldsymbol{\sigma}^2}\hat{\boldsymbol{\theta}}_t(\boldsymbol{\sigma}^2)$, and the derivation in Appendix B.1 applies by replacing $\boldsymbol{\Phi}$ with $\boldsymbol{\sigma}^2$, giving the implicit gradient in Eq. (10).

## C  Synthetic Experiments to Compare MAML, Meta-SGD and Shrinkage

In this section, we demonstrate the difference in behavior between learning the learning rate per module and learning the shrinkage prior per module on two synthetic few-shot learning problems.

In the first experiment we show that when the number of task adaptation steps in meta-training is sufficiently large for task parameters to reach convergence, meta-learning the learning rate per module has a similar effect as shrinkage regularization when evaluated at the *same* adaptation step in meta-testing; however, this does not generalize to other adaptation horizons. In the second experiment, when the required number of adaptation steps is longer than meta-training allows, the learned learning rate is determined by the local curvature of the task likelihood function whereas the learned shrinkage variance is determined by task similarity from the data generation prior. Grouping parameters with similar learned learning rates or variances then induces different "modules," which correspond to different aspects of the meta-learning problem (namely, adaptation curvature vs. task similarity).

We compare the following three algorithms:

1. vanilla MAML[13]: does not have modular modeling; learns the initialization and uses a single learning rate determined via hyper-parameter search.

2. Meta-SGD[30]: learns the initialization as well as a separate learning rate for each parameter.

3. $\boldsymbol{\sigma}$-Reptile: learns the initialization and prior variance $\sigma_m^2$ for each module.

We run each algorithm on two few-shot learning problems, both of which have the same hierarchical normal data generation process:

$$\boldsymbol{\theta}_{m,t} \sim \mathcal{N}(\boldsymbol{\theta}_{m,t}|\boldsymbol{\phi}_m, \sigma_m^2),$$
$$\mathbf{x}_{t,n} \sim \mathcal{N}(\mathbf{x}_{t,n}|\boldsymbol{\mu}(\boldsymbol{\theta}_t), \Xi),$$

for each latent variable dimension $m$, task $t$, and data point $n$. The hyper-parameters $\boldsymbol{\phi}$ and $\boldsymbol{\sigma}^2$ are shared across all tasks but unknown, and each parameter dimension $\theta_m$ has different prior variance, $\sigma_m^2$. For simplicity, we let every parameter dimension $m$ correspond to one module for Meta-SGD and $\boldsymbol{\sigma}$-Reptile. The $n$-th observation $\mathbf{x}_{t,n}$ for task $t$ is sampled from a Gaussian distribution. The mean is a known function of task parameter $\boldsymbol{\theta}$. The observation noise variance $\Xi = \mathrm{diag}(\boldsymbol{\xi}^2) = \mathrm{diag}(\xi_1^2, \dots, \xi_D^2)$ is a fixed and known diagonal matrix. The difference between the two problems is that $\boldsymbol{\mu}$ is a linear function of $\boldsymbol{\theta}_t$ in the first problem and non-linear in the second.

The task is few-shot density estimation, that is, to estimate the parameters $\boldsymbol{\theta}_{\tilde{t}}$ of a new task $\mathcal{T}_{\tilde{t}}$ given a few observations $\{\mathbf{x}_{\tilde{t},n}\}$, where $N_t^{\mathrm{train}} = N_t^{\mathrm{val}}$ for all tasks. To understand the behavior of different algorithms under different task loss landscapes we examine different transformation functions $\boldsymbol{\mu}$.

Note that the data generation process matches the Bayesian hierarchical model of shrinkage and thus $\boldsymbol{\sigma}$-Reptile may have an advantage in terms of predictive performance. However, the main purpose of this section is to demonstrate the *behavior* of these approaches and not to focus on which performs better in this simple task.

For each method, we use gradient descent for task adaptation to optimize the training loss (negative log-likelihood), and then evaluate the generalization loss on holdout tasks. MAML meta-learns the initialization of TASKADAPT, $\boldsymbol{\phi}$, while Meta-SGD meta-learns both $\boldsymbol{\phi}$ and a per-parameter learning rate $\alpha_m$, and $\boldsymbol{\sigma}$-Reptile meta-learns the shrinkage prior mean $\boldsymbol{\phi}$ and per-parameter variance $\boldsymbol{\sigma}_m^2$. Hyperparameters for each algorithm (i.e., learning rates, number of adaptation steps, and initial values of $\boldsymbol{\sigma}$) were chosen by extensive random search. We chose the values that minimized the generalization loss in meta-test. Because the model is small, we are able to run MAML and Meta-SGD for up to 200 task adaptation steps.

## C.1 Experiment 1: Linear transformation

(a) Illustration of the hierarchical normal distribution with linear $\mu$.

(b) Generalization loss in meta-testing.

Figure 7: Experiment 1: Linear transform.

We begin with a simple model – a joint normal posterior distribution over $\boldsymbol{\theta}_t$ with parameters

$$\begin{aligned} M &= 8, \\ D &= 9, \\ \boldsymbol{\phi} &= \mathbf{1}_M, \\ \boldsymbol{\sigma} &= [8, 8, 8, 8, 2, 2, 2, 2], \\ \boldsymbol{\xi} &= [8, 8, 8, 8, 5, 5, 5, 5, 1], \end{aligned}$$

and transformation

$$\boldsymbol{\mu}(\boldsymbol{\theta}_t) = [\mathbf{I}_M, \mathbf{1}_M/\sqrt{M}]^\top \boldsymbol{\theta}_t.$$

Figure 8: Mean absolute error of the estimate of each parameter as a function of task adaptation step.

The mean of the observations is thus $\theta$ in the first $M$ dimensions and $\frac{1}{\sqrt{M}}\sum_m \theta_m$ in the final dimension. To make each task nontrivial, we let $\xi$ be small in the final dimension (i.e., $\xi_M$ is small) so that the posterior of $\theta$ is restricted to a small subspace near the $\sum_m \theta_m = \frac{\sqrt{M}}{N_t^{\text{train}}}\sum_n x_{t,n}$ hyperplane. Gradient descent thus converges slowly regardless of the number of observations. Fig. 7a shows an example of this model for the first two dimensions.

Clearly, there are two distinct modules, $\theta_{1:4}$ and $\theta_{5:8}$ but, in this experiment, we do not give the module structure to the algorithms and instead treat each dimension as a separate module. This allows us to evaluate how well the algorithms can identify the module structure from data.

Note that for every task the loss function is quadratic and its Hessian matrix is constant in the parameter space. It is therefore an ideal situation for Meta-SGD to learn an optimal preconditioning matrix (or learning rate per dimension).

Fig. 7b shows the generalization loss on meta-testing tasks. With the small number of observations for each task, the main challenge in this task is overfitting. Meta-SGD and $\sigma$-Reptile obtain the same minimum generalization loss, and both are better than the non-modular MAML algorithm. Importantly, Meta-SGD reaches the minimum loss at step 95, which is the number of steps in meta-training, and then begins to overfit. In contrast, $\sigma$-Reptile does not overfit due to its learned regularization.

Fig. 8 further explains the different behavior of the three algorithms. The mean absolute error (MAE) for each of the 8 parameter dimensions is shown as a function of task adaptation step. MAML shares a single learning rate for all parameters, and thus begins to overfit in different dimensions at different steps, resulting in worse performance. Meta-SGD is able to learn two groups of learning rates, one per each ground-truth module. It learns to coordinate the two learning rates so that they reach the lowest error at the same step, after which it starts to overfit. The learning rate is limited by the curvature enforced by the last observation dimension. $\sigma$-Reptile shares a single learning rate so the error of every dimension drops at about the same speed initially and each dimension reaches its minimum at different steps. It learns two groups of variances so that all parameters are properly regularized and maintain their error once converged, instead of overfitting.

## C.2 Experiment 2: Nonlinear transform.

(a) Illustration of the hierarchical normal distribution with non-linear $\mu$.

(b) Generalization loss in meta-testing.

Figure 9: Experiment 2: Nonlinear spiral transform.

In the second experiment, we explore a more challenging optimization scenario where the mean is a nonlinear transformation of the task parameters. Specifically,

$$\begin{aligned}
M &= 10, \\
D &= 10, \\
\boldsymbol{\sigma} &= [4, 4, 4, 4, 4, 4, 4, 4, 8, 8], \\
\boldsymbol{\phi} &= 2 \cdot \mathbf{1}_M, \\
\boldsymbol{\xi} &= 10 \cdot \mathbf{1}_D.
\end{aligned}$$

The true modules are $\boldsymbol{\theta}_{1:8}$ and $\boldsymbol{\theta}_{9:10}$. The transformation $\boldsymbol{\mu}_t(\boldsymbol{\theta}_t)$ is a "swirl" effect that rotates non-overlapping pairs of consecutive parameters with an angle proportional to their $L_2$ distance from the origin. Specifically, each consecutive non-overlapping pair $(\mu_{t,d}, \mu_{t,d+1})$ is defined as

$$\begin{bmatrix} \mu_{t,d} \\ \mu_{t,d+1} \end{bmatrix} = \mathrm{Rot}\left(\omega \sqrt{\theta_{t,d}^2 + \theta_{t,d+1}^2}\right) \cdot \begin{bmatrix} \theta_{t,d} \\ \theta_{t,d+1} \end{bmatrix}, \quad \text{for } d = 1, 3, ..., M-1,$$

where

$$\mathrm{Rot}(\varphi) = \begin{bmatrix} \cos\varphi & -\sin\varphi \\ \sin\varphi & \cos\varphi \end{bmatrix}, \tag{46}$$

and $\omega = \pi/5$ is the angular velocity of the rotation. This is a nonlinear volume-preserving mapping that forms a spiral in the observation space. Fig. 9a shows an example of the prior and likelihood function in 2-dimensions of the parameter space.

Compared to the previous example, this highly nonlinear loss surface with changing curvature is more realistic. First-order optimizers are constrained by the narrow valley formed by the "swirly" likelihood function, and thus all algorithms require hundreds of adaptation steps to minimize the loss.

In this case, the best per-parameter learning rate learned by Meta-SGD is restricted by the highest curvature in the **likelihood** function along the optimization trajectory. In contrast, the optimal per-parameter variance estimated by $\boldsymbol{\sigma}$-Reptile depends on the **prior** variance $\sigma_m^2$ in the data generating process, regardless of the value of $\omega$, given that optimization eventually converges.

As a consequence, Meta-SGD and $\boldsymbol{\sigma}$-Reptile exhibit very different behaviors in their predictive performance in Fig. 9b. Meta-SGD overfits after about 700 steps, while $\boldsymbol{\sigma}$-Reptile keeps improving after 1000 steps.

Also, because MAML and Meta-SGD require backpropagation through the adaptation process, when the number of adaptations steps is higher than 100, we notice that meta-training becomes unstable. As a result, the best MAML and Meta-SGD hyperparameter choice has fewer than 100 adaptation steps in meta-training. These methods then fail to generalize to the longer optimization horizon required in this problem.

We do not show the per-parameter MAE trajectory as in the previous section because this optimization moves through a spiraling, highly coupled trajectory in the parameter space, and thus per-parameter MAE is not a good metric to measure the progress of optimization.

## D  Algorithm Implementation Details

### D.1  Implementation details for iMAML

In this work, we implement and compare our algorithm to iMAML-GD — the version of iMAML that uses gradient descent within task adaptation [3, Sec. 4] — as this better matches the proximal gradient descent optimizer used in our shrinkage algorithms.

In our implementation of conjugate gradient descent, to approximate the inverse Hessian we apply a damping term following Rajeswaran et al. [3] and restrict the number of conjugate gradient steps to 5 for numerical stability. The meta update is then

$$\Delta_t^{\mathrm{iMAML}} = \left((1+d)\mathbf{I} + \tfrac{1}{\lambda}\nabla_{\boldsymbol{\theta}_t}^2 \ell_t^{\mathrm{train}}(\boldsymbol{\theta}_t)\right)^{-1} \nabla_{\boldsymbol{\theta}_t} \ell_t^{\mathrm{val}}(\boldsymbol{\theta}_t), \tag{47}$$

where $d$ is the damping coefficient. We treat $d$ and the number of conjugate gradient steps as hyperparameters and choose values using hyperparameter search.

## D.2 Implementation details for shrinkage prior algorithms

As in iMAML, we apply damping to the Hessian matrix when running conjugate gradient descent to approximate the product $\nabla_{\boldsymbol{\theta}_t}\ell_t^{\mathrm{val}}(\hat{\boldsymbol{\theta}}_t)\mathbf{H}_{\hat{\boldsymbol{\theta}}_t\hat{\boldsymbol{\theta}}_t}^{-1}$ in $\boldsymbol{\sigma}$-iMAML (Eq. 7) and $\boldsymbol{\sigma}$-Reptile (Eq. 10),

$$\mathbf{H}_{\hat{\boldsymbol{\theta}}_t\hat{\boldsymbol{\theta}}_t} = -\tilde{d}\mathbf{I} - \boldsymbol{\Sigma}^{-1} - \nabla_{\hat{\boldsymbol{\theta}}_t\hat{\boldsymbol{\theta}}_t}\log p\left(\mathcal{D}_t^{\mathrm{train}}|\boldsymbol{\theta}_t\right), \tag{48}$$

where $\boldsymbol{\Sigma} = \mathrm{Diag}(\sigma_1^2\mathbf{I}_{D_1}, \sigma_2^2\mathbf{I}_{D_2}, \dots, \sigma_M^2\mathbf{I}_{D_M})$, and $\mathrm{Diag}(\dots, \mathbf{B}_m, \dots)$ denotes a block diagonal matrix with $m$-th block $\mathbf{B}_m$. Note that the damped update rule reduces to that of iMAML when $\sigma_m^2 = 1/\lambda, \forall m$ and $\tilde{d} = d\lambda$.

Additionally, we apply a diagonal pre-conditioning matrix in the same structure as $\boldsymbol{\Sigma}$, $\mathbf{P} = \mathrm{Diag}(p_1\mathbf{I}_{D_1}, p_2\mathbf{I}_{D_2}, \dots, p_M\mathbf{I}_{D_M})$ with $p_m = \max\{\sigma_m^{-2}/10^3, 1\}$ to prevent an ill-conditioned Hessian matrix when the prior variance becomes small (strong prior).

We also clip the value of $\sigma_m^2$ to be in $[10^{-5}, 10^5]$. This clipping improves the stability of meta learning, and the range is large enough to not affect the MAP estimate of $\hat{\boldsymbol{\theta}}_t$.

Finally, we incorporate a weak regularizer on the shrinkage variance to encourage sparsity in the discovered adaptable modules. The regularized objective for learning $\boldsymbol{\sigma}^2$ becomes

$$\frac{1}{T}\ell_{\mathrm{PLL}} + \beta\log\mathrm{IG}(\boldsymbol{\sigma}^2), \tag{49}$$

where IG is the inverse Gamma distribution with shape $\alpha = 1$ and scale $\beta$. Unless otherwise stated, we use $\beta = 10^{-5}$ for sinusoid and image experiments, and $\beta = 10^{-7}$ for text-to-speech experiments. We find that this regularization simply reduces the learned $\sigma_m^2$ of irrelevant modules without affecting generalization performance.

## D.3 Proximal Gradient Descent and Proximal Adam with L2 regularization

The pseudo-code of Proximal Gradient Descent [45] with an L2 regularization is presented in Algorithm 2. We also modify the Adam optimizer [46] to be a proximal method and present the pseudo-code in Algorithm 3.

| **Algorithm 2:** Proximal Gradient Descent with L2 Regularization. | **Algorithm 3:** Proximal Adam with L2 Regularization. |
|---|---|
| **Input:** Parameter $\boldsymbol{\theta}_t$, gradient $\mathbf{g}_t$, step size $\alpha_t$, regularization center $\phi$, L2 regularization scale $\lambda$. | **Input:** Parameter $\boldsymbol{\theta}_t$, gradient $\mathbf{g}_t$, step size $\alpha_t$, regularization center $\phi$, L2 regularization scale $\lambda$, $\epsilon$ for Adam. |
| $\boldsymbol{\theta}_{t+\frac{1}{2}} = \boldsymbol{\theta}_t - \alpha_t\mathbf{g}_t$ <br> $\boldsymbol{\theta}_{t+1} = (\boldsymbol{\theta}_{t+\frac{1}{2}} - \phi)/(1 + \lambda\alpha_t) + \phi$ <br> **return** $\boldsymbol{\theta}_{t+1}$ | $\boldsymbol{\theta}_{t+\frac{1}{2}}, \hat{\mathbf{v}}_{t+1} = \mathrm{Adam}(\boldsymbol{\theta}_t, \mathbf{g}_t, \alpha_t)$ <br> $\boldsymbol{\theta}_{t+1} = (\boldsymbol{\theta}_{t+\frac{1}{2}} - \phi)/(1 + \lambda\alpha_t/\sqrt{\hat{\mathbf{v}}_{t+1} + \epsilon}) + \phi$ <br> **return** $\boldsymbol{\theta}_{t+1}$ |

# E  Experiment Details and Additional Short Adaptation Experiments

In our experiments, we treat each layer (e.g., the weights and bias for a convolutional layer are a single layer) as a module, including batch-normalization layers which we adapt as in previous work [13]. Other choices of modules are straightforward but we leave exploring these for future work. For the chosen hyperparameters, we perform a random search for each experiment and choose the setting that performs best on validation data.

## E.1 Augmented Omniglot experiment details

We follow the many-shot Omniglot protocol of Flennerhag et al. [4], which takes the 46 Omniglot alphabets that have 20 or more character classes and creates one 20-way classification task for each alphabet by sampling 20 character classes from that alphabet. These classes are then kept fixed for the duration of the experiment. Of the 46 alphabets, 10 are set aside as a held-out test set, and the

remainder are split between train and validation. The assignments of alphabets to splits and of classes to tasks is determined by a random seed at the beginning of each experiment. For each character class, there are 20 images in total, 15 of which are set aside for training (i.e., task adaptation) and 5 for validation. This split is kept consistent across all experiments. All Omniglot images are downsampled to $28 \times 28$. Note that this protocol differs significantly from the standard few-shot Omniglot protocol (discussed below), where each task is created by selecting $N$ different characters (from any alphabet), randomly rotating each character by $\{0, 90, 180, 270\}$ degrees, and then randomly selecting $K$ image instances of that (rotated) character.

At each step of task adaptation and when computing the validation loss, a batch of images is sampled from the task. Each of these images is randomly augmented by re-scaling it by a factor sampled from $[0.8, 1.2]$, and translating it by a factor sampled from $[-0.2, 0.2]$. In the large-data regime, images are also randomly rotated by an angle sampled from $\{0, \ldots, 359\}$ degrees. In the small-data regime, no rotation is applied.

We use the same convolutional network architecture as Flennerhag et al. [4], which differs slightly from the network used for few-shot Omniglot (detailed below). Specifically, the few-shot Omniglot architecture employs convolutions with a stride of 2 with no max pooling, whereas the architecture for many-shot Omniglot uses a stride of 1 with $2 \times 2$ max pooling. In detail, the architecture for many-shot Omniglot consists of 4 convolutional blocks, each made up of a $3 \times 3$ convolutional layer with 64 filters, a batch-normalization layer, a ReLU activation, and a $2 \times 2$ max-pooling layer, in that order. The output of the final convolutional block is fed into an output linear layer and then a cross-entropy loss.

Table 4: Hyperparameters for the large-data augmented Omniglot classification experiment. Chosen with random search on validation task set.

|  | Reptile | iMAML | $\sigma$-Reptile | $\sigma$-iMAML |
|---|---|---|---|---|
| **Meta-training** | | | | |
| Meta optimizer | SGD | Adam | Adam | Adam |
| Meta learning rate ($\phi$) | 1.2 | 1.8e-3 | 6.2e-3 | 5.4e-3 |
| Meta learning rate ($\log \sigma^2$) | - | - | 1.6e-2 | 0.5 |
| Meta training steps | 5k | 5k | 5k | 5k |
| Meta batch size (# tasks) | 20 | 20 | 20 | 20 |
| Damping coefficient | - | 0.1 | 0.16 | 9e-2 |
| Conjugate gradient steps | - | 1 | 4 | 5 |
| **Task adaptation (adaptation step for meta-test is in parentheses)** | | | | |
| Task optimizer | Adam | ProximalGD | ProximalGD | ProximalGD |
| Task learning rate | 9.4e-3 | 0.5 | 0.52 | 0.37 |
| Task adaptation steps | 100 (100) | 100 (100) | 100 (100) | 100 (100) |
| Task batch size (# images) | 20 | 20 | 20 | 20 |

### E.1.1 Large-data regime

In the large-data regime, we use 30 alphabets for training, each with 15 image instances per character class. Hyperparameters for each algorithm in this domain are shown in Table 4. For iMAML, we use $\lambda = 1.3e{-}4$.

Fig. 2 shows the learned variances and resulting test accuracies when adapting different modules. We find in this dataset all the shrinkage algorithms choose the last linear layer for adaptation, and the performance matches that of adapting all layers with learned variance within a $95\%$ confidence interval of 0.08.

Table 5: Hyperparameters for the small-data augmented Omniglot experiment. Chosen with random search on validation task set.

|  | Reptile | iMAML | $\sigma$-Reptile | $\sigma$-iMAML |
|---|---|---|---|---|
| **Meta-training** | | | | |
| Meta optimizer | SGD | Adam | Adam | Adam |
| Meta learning rate ($\phi$) | 0.1 | 1e-3 | 1.4e-4 | 2e-4 |
| Meta learning rate ($\log \sigma^2$) | - | - | 5e-3 | 1.3 |
| Meta training steps | 1000 | 1000 | 1000 | 1000 |
| Meta batch size (# tasks) | 20 | 20 | 20 | 20 |
| Damping coefficient | - | 0.5 | 1e-3 | 2e-3 |
| Conjugate gradient steps | - | 2 | 5 | 3 |
| Regularization scale ($\beta$) | - | - | 1e-6 | 1e-7 |
| **Task adaptation** | | | | |
| Task optimizer | Adam | ProximalGD | ProximalGD | ProximalGD |
| Task learning rate | 4e-3 | 0.4 | *(see Table 6)* | *(see Table 6)* |
| Task adaptation steps | 100 | 100 | 100 | 100 |
| Task batch size (# images) | 20 | 20 | 20 | 20 |

Table 6: Task optimizer learning rate for different numbers of training instances per character class in small-data augmented Omniglot.

| # train instances | 1 | 3 | 5 | 10 | 15 |
|---|---|---|---|---|---|
| $\sigma$-Reptile task LR | 0.4 | 0.31 | 0.22 | 0.12 | 0.03 |
| $\sigma$-iMAML task LR | 0.25 | 0.17 | 0.14 | 0.1 | 0.05 |

### E.1.2 Small-data regime

In the small-data regime, we evaluate the performance of Reptile, iMAML, $\sigma$-Reptile, and $\sigma$-iMAML across variants of the many-shot augmented Omniglot task with different numbers of training alphabets and training image instances per character class, without rotation augmentation. We train each algorithm 10 times with each of $5, 10, 15$, and $20$ training alphabets and $1, 3, 5, 10$, and $15$ training instances per character class.

We meta-train all algorithms for 1000 steps using 100 steps of adaptation per task. We use a meta-batch size of 20, meaning that the same task can appear multiple times within a batch, although the images will be different in each task instance due to data augmentation. At meta-evaluation time, we again perform 100 steps of task adaptation on the task-train instances of each test alphabet, and report the accuracy on the task-validation instances.

Hyperparameters for each of the 4 algorithms that we evaluate on small-data augmented Omniglot are listed in Table 5. These were chosen based on a comprehensive hyperparameter search on separate validation data. For Reptile, we use a linear learning rate decay that reaches $0$ at the end of meta-training, as in Nichol et al. [2]. For iMAML, we use an L2 regularization scale of $\lambda = 1/\sigma^2 = 1\mathrm{e}{-3}$. For $\sigma$-iMAML and $\sigma$-Reptile, we use a different task optimizer learning rate for different numbers of instances. These learning rates are presented in Table 6.

Due to the space limit, we show results with 1 training instance per character class and results with 15 training alphabets in Fig. 4 of the main text. The full results of the 20 experimental conditions are presented in Figs. 10 and 11, which respectively show the performance of each method as the number of training instances and alphabets vary. As discussed in the main text, each shrinkage variant outperforms or matches its corresponding non-shrinkage variant in nearly every condition. Improvements by the shrinkage variants are more consistent and pronounced when the amount of training data is limited.

(a) 1 image instance per character.

(b) 3 image instance per character.

(c) 5 image instance per character.

(d) 10 image instance per character.

(e) 15 image instance per character.

Figure 10: Test accuracy on augmented Omniglot in the small-data regime as a function of the number of training alphabets. In each plot, the number of instances per class is fixed. Each data point is the average of 10 runs and 95% confidence intervals are shown.

## E.2   Few-shot Text-to-Speech experiment details

For few-shot text-to-speech synthesis, various works [6, 1, 7, 8] have made use of speaker-encoding networks or trainable speaker embedding vectors to adapt to a new voice based on a small amount of speech data. These works achieved success to some extent when there were a few training utterances, but the performance saturated quickly beyond 10 utterances [7] due to the bottleneck in the speaker specific components. Arik et al. [6] and Chen et al. [1] found that the performance kept improving with more utterances by fine-tuning the entire TTS model, but the adaptation process had to be terminated early to prevent overfitting. As such, some modules may still be underfit while others have begun to overfit, similar to the behavior seen for MAML in Fig. 8.

In this paper, we examine the advantages of shrinkage for a WaveNet model [21]. The same method applies to other TTS architectures as well. In preliminary experiments, we found iMAML and $\sigma$-iMAML meta-learn much more slowly than Reptile and $\sigma$-Reptile. We conjecture that this is because iMAML and $\sigma$-iMAML compute meta-gradients based only on the validation data from the last mini-batch of task adaptation. In contrast, the meta update for $\phi$ from Reptile and $\sigma$-Reptile accumulates the task parameter updates computed from training mini-batches through the adaptation process. With a task adaptation horizon of 100 steps, this leads to significantly different data efficiencies. As a result, we only evaluate Reptile and $\sigma$-Reptile for this experiment.

The WaveNet model is an augoregressive generative model. At every step, it takes the sequence of waveform samples generated up to that step, and the concatenated fundamental frequency (f0) and linguistic feature sequences as inputs, and predicts the sample at the next step. The sequence of fundamental frequency controls the dynamics of the pitch in an utterance. The short-time frequency is important for WaveNet to predict low level sinusoid-like waveform features. The linguistic features encode the sequence of phonemes from text. They are used by WaveNet to generate speech with

(a) 5 training alphabets.

(b) 10 training alphabets.

(c) 15 training alphabets.

(d) 20 training alphabets.

Figure 11: Test accuracy on augmented Omniglot in the small-data regime as a function of the number of training instances. In each plot, the number of training alphabets is fixed. Each data point is the average of 10 runs and 95% confidence intervals are shown.

Figure 12: Architecture of the multi-speaker WaveNet model. The single-speaker WaveNet model used by Reptile and $\sigma$-Reptile, does not include the speaker embedding lookup table and the corresponding conditional input layer.

corresponding content. The dynamics of the fundamental frequency in f0 together with the phoneme duration contained in the linguistic feature sequence contains important information about the prosody of an utterance (word speed, pauses, emphasis, emotion, etc), which change at a much slower time-scale. While the fundamental frequency and prosody in the inputs contain some information about the speaker identity, the vocal tract characteristics of a voice that is unique to each speaker cannot be inferred from the inputs, and has to be learned by the WaveNet model from waveform samples through task adaptation.

The full architecture of a multi-speaker WaveNet model used by SEA-Emb and SEA-All in Chen et al. [1] is shown in Fig. 12. For Reptile and $\sigma$-Reptile, we use a single-speaker model architecture that excludes the speaker embedding lookup table and associated conditional input layers in the residual blocks. The single-speaker model is comprised of one input convolutional layer with $1 \times 1$ kernel, one input skip-output $1 \times 1$ layer, 30 residual of dilated causal-convolutional blocks (each including

4 layers), and 2 $1 \times 1$ output layers before feeding into a 256-way softmax layer. We treat every layer as a module, for a total of 123 modules (the output layer of the last block is not used for prediction).

To speed up meta-learning, we first pretrain a single-speaker model with all training speaker data mixed, and initialize $\sigma$-Reptile and Reptile with that pretrained model. This is reminiscent of other meta-learning works for few-shot image classification that use a pretrained multi-head model to warm-start when the backbone model (e.g. ResNet) is large. This pretrained TTS model learns to generate speech-like samples but does not maintain a consistent voice through a single sentence In contrast to other meta-learning works that fix the feature extracting layers, we then meta-learn all WaveNet layers to identify most task-specific modules.

We run the multispeaker training and pre-training for meta-learning methods for 1M steps on a proprietary dataset of 61 speakers each with 2 hours of speech data with 8-bit encoding sampled at 24K frequency. For $\sigma$-Reptile and Reptile, we further run meta-learning for 8K meta-steps with 100 task adaptation steps in the inner loop for the same set of speakers. Each task consists of 100 utterances (roughly 8 minutes of speech). We evaluate the model on 10 holdout speakers, in two data settings, with 100 utterances or 50 utterances (roughly 4 minutes of speech) per task. We run up to 10,000 task adaptation steps in meta-test. SEA-All and Reptile both finetune all parameters. They overfit quickly in meta-testing after about 1,500 and 3,000 adaptation steps with 4-min and 8-min of data, respectively. We therefore early terminate task adaptation for these algorithms to prevent overfitting.

To measure the sample quality of naturalness, we have human evaluators rate each sample on a five-point Likert Scale (1: Bad, 2: Poor, 3: Fair, 4: Good, 5: Excellent) and then we compute the mean opinion score (MOS). This is the standard approach to evaluate the quality of TTS models.

Voice similarity is computed as follows. We use a pretrained speaker verification model [47] that outputs an embedding vector, $d(x)$, known as a $d$-vector, for each utterance $x$. We first compute the mean of $d$-vectors from real utterances of each test speaker, $t$. $\bar{d}_t := \sum_n d(x_{t,n})/N_t$. Given a model adapted to speaker $t$, we compute the sample similarity for every sample utterance $x_i$ as

$$\text{sim}(x_i, t) = \cos(d(x_i), \bar{d}_t).$$

### E.3 Additional short adaptation experiment: sinusoid regression

We follow the standard sinusoid regression protocol of Finn et al. [13] in which each task consists of regressing input to output of a sinusoid $y = a\sin(x - b)$ uniformly sampled with amplitude $a \in [0.1, 5]$ and phase $b \in [0, \pi]$. Each task is constructed by sampling 10 labelled data points from input range $x \in [-5, 5]$. We learn a regression function with a 2-layer neural network with 40 hidden units and ReLU nonlinearities and optimise the mean-squared error (MSE) between predictions and true output values.

Our method is agnostic to the choice of modules. For this small model, consider each set of network parameters as a separate module. In total, we define 6 modules: $\{b_i, w_i\}$, for $i = 0, 1, 2$, where $b_i$ and $w_i$ denote the bias and weights of each layer. We run each shrinkage variant for 100K meta-training steps, and evaluate the generalization error on 100 holdout tasks. The hyperparameters of all three shrinkage algorithm variants are given in Table 7.

We show the learned variance from $\sigma$-MAML, $\sigma$-iMAML and $\sigma$-Reptile in Fig. 13(a,d,g) respectively. In all experiments, we observe that the learned variances $\sigma_m^2$ for the first 2 modules $(b_0, w_0)$ are significantly larger than the rest. This implies that our method discovers that these modules are task-specific and should change during adaptation whereas the other modules are task-independent.

To confirm that the learned variances correspond to task-specificity, we adapt one layer at a time on holdout tasks and keep the other layers fixed. Fig. 13 shows that adapting only the discovered first layer results in both low error and accurately reconstructed sinusoids, whereas adapting the other modules does not.

### E.4 Additional short adaptation experiment: few-shot image classification

We next look at two standard benchmarks for few-shot image classification, Omniglot and *mini*ImageNet, which perform task adaptation in meta-training for up to 20 steps.

Table 7: Hyperparameters for the few-shot sinusoid regression experiment. Chosen with random search on validation task set.

| | $\sigma$-MAML | $\sigma$-Reptile | $\sigma$-iMAML |
|---|---|---|---|
| **Meta-training** | | | |
| Meta optimizer | Adam | Adam | Adam |
| Meta learning rate ($\phi$) | 9.8e-4 | 3.0e-3 | 5.7e-3 |
| Meta learning rate ($\log \boldsymbol{\sigma}^2$) | 4.4e-3 | 1.4e-4 | 1.8e-3 |
| Meta training steps | 100k | 100k | 100k |
| Meta batch size (# tasks) | 5 | 5 | 5 |
| Damping coefficient | - | 6e-3 | 0.5 |
| Conjugate gradient steps | - | 7 | 1 |
| **Task adaptation** | | | |
| Task optimizer | ProximalGD | ProximalGD | ProximalGD |
| Task learning rate | 8.3e-4 | 1.4e-4 | 3.9e-4 |
| Task adaptation steps | 68 | 100 | 100 |
| Task batch size (# data points) | 10 | 10 | 10 |

(a) Learned $\boldsymbol{\sigma}$ for each module ($\sigma$-MAML).

(b) MSE under different adaptation schemes.

(c) Prediction after 60 steps of adaptation.

(d) Learned $\boldsymbol{\sigma}$ for each module ($\sigma$-iMAML).

(e) MSE under different adaptation schemes.

(f) Prediction after 60 steps of adaptation.

(g) Learned $\sigma$ for each module ($\sigma$-Reptile).

(h) MSE under different adaptation schemes.

(i) Prediction after 60 steps of adaptation.

Figure 13: Sinusoid regression with $\sigma$-MAML (top row), $\sigma$-iMAML (middle row), and $\sigma$-Reptile (bottom row). In the left column (a,d,g) we show the learned $\sigma_m$ for each module. In the middle column (b,e,h) we show the mean squared error, averaged over 100 tasks, as a function of task adaptation step, while adapting only a single module (the dashed lines) or using the learned $\boldsymbol{\sigma}$ (pink). Finally, the right column (c,f,i) shows predictions under each adaptation scheme. Note that the model trained using the learned $\boldsymbol{\sigma}$ (pink) overlaps with the model with the first layer adapted (dark blue) in (b–c,e–f,h–i).

### E.4.1 Few-shot Omniglot

The Omniglot dataset [48, 49] consists of 20 samples of 1623 characters from 50 different alphabets. The dataset is augmented by creating new characters that are rotations of each of the existing characters by $0, 90, 180$, or $270$ degrees. We follow the standard $N$-way $K$-shot classification setting where a task is generated by randomly sampling $N$ characters and training the model on $K$ instances of each [49–51]. Character classes are partitioned into separate meta-train and meta-test splits, and the instances (images) of each character are also split into separate (task) train and (task) validation subsets.

We use the same 4-block convolutional architecture as in Finn et al. [13]. This architecture consists of 4 convolutional blocks, each made up of a $3 \times 3$ convolutional layer with 64 filters and stride 2, a batch-normalization layer [52], and a ReLU activation, in that order. The output of the final convolutional block is fed into an output linear layer and a softmax, and trained with the cross-entropy loss. All images are downsampled to $28 \times 28$.

Hyperparameters for the six algorithms evaluated in this experiment are presented in Table 8. Module discovery and classification performance for this dataset are discussed below.

Table 8: Hyperparameters for the few-shot Omniglot classification. Chosen with random search on the validation task set.

|  | MAML | Reptile | iMAML | $\sigma$-MAML | $\sigma$-Reptile | $\sigma$-iMAML |
|---|---|---|---|---|---|---|
| Meta-training | | | | | | |
| Meta optimizer | Adam | SGD | Adam | Adam | SGD | Adam |
| Meta learning rate ($\phi$) | 4.8e-3 | 1.8 | 1e-3 | 1.7e-4 | 1.6 | 7.8e-3 |
| Meta learning rate ($\log \boldsymbol{\sigma}^2$) | - | - | - | 7.4e-3 | 7e-4 | 4e-3 |
| Meta training steps | 60k | 100k | 60k | 60k | 100k | 60K |
| Meta batch size (# tasks) | 32 | 5 | 32 | 32 | 5 | 32 |
| Damping coefficient | - | - | 1.0 | - | 6e-3 | 0.18, |
| Conjugate gradient steps | - | - | 4 | - | 1 | 2 |
| Task adaptation (adaptation step and batch size for meta-test are in parentheses) | | | | | | |
| Task optimizer | SGD | Adam | ProximalGD | ProximalGD | Adam | ProximalGD |
| Task learning rate | 0.968 | 8e-4 | 0.23 | 0.9 | 6e-4 | 1.3 |
| Task adaptation steps | 1 (50) | 5 (50) | 19 (50) | 3 (50) | 8 (50) | 8 (50) |
| Task batch size (# images) | 5 (5) | 10 (5) | 5 (5) | 5 (5) | 10 (5) | 5 (5) |

### E.4.2 *mini*ImageNet

The *mini*ImageNet dataset [50, 51] is, as the name implies, a smaller and easier many-task, few-shot variant of the ImageNet dataset. However, its images are larger and more challenging than those of Omniglot. *mini*ImageNet consists of 100 classes (64 train, 12 validation, and 24 test) with images downsampled to $84 \times 84$. We follow the standard *mini*ImageNet protocol and train in the $N$-way $K$-shot paradigm using the same 4-block convolutional architecture as in previous work [50, 13]. Each convolutional block consists of a $3 \times 3$ convolutional layer with 32 filters, a batch-normalization layer [52], and a ReLU activation, in that order. As in Omniglot, the output of the final convolutional block is fed into an output linear layer and a softmax, and trained with the cross-entropy loss.

Hyperparameters for each algorithm on *mini*ImageNet are presented in Table 9. For iMAML, we use $\lambda = 0.14$. Module discovery and classification performance for this dataset are discussed below.

### E.4.3 Module discovery

We first discuss module discovery for these two datasets and then compare classification accuracies in the next section.

Fig. 14 and Fig. 15 present our module discovery results for all three shrinkage algorithms on few-shot Omniglot and *mini*ImageNet. We see that after training there is always one layer that has a learned variance that is much larger than all other layers. As shown in Fig. 14(d-e) and Fig. 15(d-e), we observe that by adapting only this task-specific module, the model is able to achieve high accuracy at

Table 9: Hyperparameters for the few-shot *mini*ImageNet classification experiment. Chosen with random search on validation task set.

| | MAML | Reptile | iMAML | $\sigma$-MAML | $\sigma$-Reptile | $\sigma$-iMAML |
|---|---|---|---|---|---|---|
| **Meta-training** | | | | | | |
| Meta optimizer | Adam | SGD | Adam | Adam | Adam | Adam |
| Meta learning rate ($\phi$) | 1e-3 | 0.45 | 2.5e-4 | 1e-3 | 6e-4 | 2e-4 |
| Meta learning rate ($\log \boldsymbol{\sigma}^2$) | - | - | - | 5e-2 | 2.5e-2 | 0.2 |
| Meta training steps | 60k | 100k | 60k | 60k | 100k | 60k |
| Meta batch size (# tasks) | 4 | 5 | 4 | 5 | 3 | 5 |
| Damping coefficient | - | - | 6.5e-2 | - | 1e-2 | 2e-3 |
| Conjugate gradient steps | - | - | 5 | - | 3 | 2 |
| **Task adaptation (adaptation step and batch size for meta-test are in parentheses)** | | | | | | |
| Task optimizer | SGD | Adam | ProximalGD | ProximalGD | ProximalGD | ProximalGD |
| Task learning rate | 1e-2 | 1.5e-3 | 4e-2 | 0.1 | 1.35 | 1.6 |
| Task adaptation steps | 5 (50) | 5 (50) | 17 (50) | 5 (50) | 7 (50) | 1 (50) |
| Task batch size (# images) | 5 (5) | 10 (5) | 5 (5) | 5 (5) | 10 (5) | 5 (5) |

(a) Learned $\sigma^2$ for $\boldsymbol{\sigma}$-MAML.   (b) Learned $\sigma^2$, $\boldsymbol{\sigma}$-iMAML.   (c) Learned $\sigma^2$ for $\boldsymbol{\sigma}$-Reptile.

(d) Accuracy for $\boldsymbol{\sigma}$-MAML.   (e) Accuracy, $\boldsymbol{\sigma}$-iMAML.   (f) Accuracy for $\boldsymbol{\sigma}$-Reptile.

Figure 14: Learned variances and test accuracies on Omniglot. (a) & (b) & (c) show the learned variance per module with $\boldsymbol{\sigma}$-MAML, $\boldsymbol{\sigma}$-iMAML and $\boldsymbol{\sigma}$-Reptile, respectively. (d) & (e) & (f) show the average test accuracy at the end of task adaptation with $\boldsymbol{\sigma}$-MAML, $\boldsymbol{\sigma}$-iMAML and $\boldsymbol{\sigma}$-Reptile. Each bar shows the accuracy after task adaptation either with all layers frozen except one. Colors map to the colors of (a) & (b) & (c). $\mathrm{bn}_i$ and $\mathrm{conv}_i$ denote $i$-th batch normalization layer and convolutional layer, and $\mathrm{output}$ is the linear output layer. The pink dashed line shows the accuracy after adaptation with the learned $\sigma$ and the black dashed line is the chance accuracy.

test time, equal to the performance achieved when adapting all layers according to the learned $\boldsymbol{\sigma}^2$. Conversely, adapting only the task-independent modules leads to poor performance.

Importantly, it is always one of the final two output layers that has the highest learned variance, meaning that these two layers are the most task-specific. This corroborates the conventional belief in image-classification convnets that output layers are more task-specific while input layers are more general, which is also validated in other meta-learning works [17, 25].

However, in most of the subfigures in Fig. 14 and Fig. 15, it is actually the *penultimate* layer that is most task-specific and should be adapted. The only algorithm for which the penultimate layer was not the most task-specific was $\boldsymbol{\sigma}$-iMAML in the Omniglot experiment. To study this further, we run $\boldsymbol{\sigma}$-iMAML on Omniglot repeatedly with a random initialization and hyper-parameter settings and keep the learned models that achieve at least $95\%$ test accuracy. In Fig. 16, we show the learned $\sigma_m^2$ of the final convolutional layer ($conv_3$) and the linear output layer ($output$) from those runs. In almost

(a) Learned $\sigma^2$ for $\boldsymbol{\sigma}$-MAML.  (b) Learned $\sigma^2$, $\boldsymbol{\sigma}$-iMAML.  (c) Learned $\sigma^2$ for $\boldsymbol{\sigma}$-Reptile.

(d) Accuracy for $\boldsymbol{\sigma}$-MAML.  (e) Accuracy, $\boldsymbol{\sigma}$-iMAML.  (f) Accuracy for $\boldsymbol{\sigma}$-Reptile.

Figure 15: Learned variances and test accuracies on *mini*ImageNet. (a) & (b) & (c) show the learned variance per module with $\boldsymbol{\sigma}$-MAML, $\boldsymbol{\sigma}$-iMAML and $\boldsymbol{\sigma}$-Reptile, respectively. (d) & (e) & (f) show the average test accuracy during at the end of task adaptation with $\boldsymbol{\sigma}$-MAML, $\boldsymbol{\sigma}$-iMAML and $\boldsymbol{\sigma}$-Reptile. Each bar shows the accuracy after task adaptation either with all layers frozen except one. Colors map to the colors of (a) & (b) & (c). $\mathrm{bn}_i$ and $\mathrm{conv}_i$ denote $i$-th batch normalization layer and convolutional layer, and output is the linear output layer. The pink dashed line shows the accuracy after adaptation with the learned $\sigma$ and the black dashed line is the chance accuracy.

Figure 16: Learned variances of the last conv layer vs output linear layer with $\boldsymbol{\sigma}$-iMAML from multiple runs.

Table 10: Test accuracy on few-shot Omniglot and few-shot *mini*ImageNet. For each algorithm, we report the mean and $95\%$ confidence interval over $10$ different runs. For each pair of corresponding methods, we bold the entry if there exists a statistically significant difference.

|  | Omniglot $N5\ K1$ | *mini*ImageNet $N5\ K1$ |
|---|---|---|
| $\boldsymbol{\sigma}$-MAML | $98.8 \pm 0.4\%$ | $\mathbf{47.7 \pm 0.5\%}$ |
| MAML | $98.7 \pm 0.7\%$ | $46.1 \pm 0.8\%$ |
| $\boldsymbol{\sigma}$-iMAML | $97.2 \pm 0.8\%$ | $47.6 \pm 1.1\%$ |
| iMAML | $97.2 \pm 1.2\%$ | $47.2 \pm 1.4\%$ |
| $\boldsymbol{\sigma}$-Reptile | $\mathbf{97.8 \pm 0.5\%}$ | $47.0 \pm 0.9\%$ |
| Reptile | $96.9 \pm 0.6\%$ | $47.4 \pm 0.9\%$ |

every case, $conv_3$ dominates *output* by an order of magnitude, meaning that the selection of the output layer in Fig. 14 was a rare event due to randomness. Adapting $conv_3$ remains the most stable choice, and most runs that adapt only it achieve a test accuracy that is not statistically significantly different from the result in Fig. 14(b). Our results extend the results of the above meta-learning works that focus only on adapting the final output layer, and match a recent independent observation in Arnold et al. [19].

Considering the different task-specific modules discovered in augmented Omniglot, TTS, sinusoid regression, and these two short-adaptation datasets, it is clearly quite challenging to hand-select task-specific modules *a priori*. Instead, meta-learning a shrinkage prior provides a flexible and general method for identifying the task-specific and task-independent modules for the domain at hand.

**Learning a prior for a single module.**
We also tried learning a single shared prior $\sigma^2$ for all variables. When using a single module like this, $\sigma^2$ grows steadily larger during meta-learning, and the shrinkage algorithms simply reduce to their corresponding standard meta-learning algorithms without the shrinkage prior. This highlights the necessity of learning different priors for different components of a model.

This observation differs from the results of Rajeswaran et al. [3], which reported an optimal value of $\lambda = 1/\sigma^2 = 2$ for iMAML. One possible explanation for this difference is that too small of a value for $\lambda$ leads to instability in computing the implicit gradient, which hinders learning. Also, when searching for the optimal $\lambda$ as a hyperparameter the dependence of the validation loss on $\lambda$ becomes less clear when $\lambda$ is sufficiently small. In our few-shot Omniglot experiments, we find that the best value of $\lambda$ for iMAML is $\lambda = 0.025$.

### E.4.4 Classification accuracy

We compare the predictive performance of all methods on held-out tasks in Table 10. While many of the results are not statistically different, the shrinkage variants show a modest but consistent improvement over the non-shrinkage algorithms. Further, $\sigma$-MAML does significantly outperform MAML on *mini*ImageNet and $\sigma$-Reptile does significantly outperform Reptile in Omniglot.

Note that we do not in general expect the shrinkage variants to significantly outperform their counterparts here because the latter do not exhibit overfitting issues on these datasets and have comparable or better accuracy than other Bayesian methods in the literature [26, 22].