[Reviews · NeurIPS 2020]

Review 1

Summary and Contributions: Within the meta-learning setting, this paper proposes a method based on Bayesian shrinkage to decide which modules to update for a new task. The key idea is to model the parameters as a hierarchical Bayesian model with each module weights modeled by a Gaussian of meta-trained mean and scalar variance. Two methods based on iMAML and Reptile are proposed, leading to slightly better generalization results, but, notably, adapting only a subset of the parameters.

Strengths: - A key motivation of this work is that we do not know a priori which modules to make adaptable and which not. This is very well supported by different datasets requiring modules at different "heights" to be adapted; the TTS model adapting the middle layers is specially nice. - The modular approach both for data-efficiency and memory constraints has clear motivations. - The paper is well-written with clear diverse evaluations

Weaknesses: - Except for 8-min TTS and Omniglot without augmentation and few instances, the results are all within statistical confidence of baselines. It would also be good to bolden those baselines that are statistically indistinguishable (and, in particular, boldening Reptile-4min in fig 5, which seems to beat \sigma-Reptile. If there bolden means "Ours" I would bolden the legend instead). - It would be useful to have more motivation/explanation of when we need long adaptation despite having few data.

Correctness: - One issue with correctness, although I don't see an easy/quick fix, is that the training does not match the intended testing conditions (assuming I fully understood it). In particular, during training all sigmas are >0, even if some are small. Then at deployment we change some of them to 0, a situation for which we have never trained. That this procedure works is not justified. For instance, it could be that some modules have small sigma because they only need small changes to have a big impact. If that were the case, this pruning would fail. - A related issue that is not clear to me is how to decide the thresohld sigma or if sigma should be the criteria for pruning (for instance, a smaller sigma in a much bigger module will have a larger norm impact on the weights, but maybe not on the function). - It's not obvious to me that sigma^2 should be the quantity being plotted instead of sigma, since that is the scale of the allowed change. Obviously it's a monotonic transformation, but the effect on the qualitative outlook of how sparse the plots are is probably substantial. - Meta-learning methods have a complex relationship with BatchNorm (see TaskNorm in ICML 2020 https://proceedings.icml.cc/static/paper_files/icml/2020/2696-Paper.pdf). How do these novel variants approach the BN layers? If I understood Fig2 correctly the parameters within BN are treated as a module, and thus have its own sigma, which makes sense. However, are we allowing running stats to adapt?

Clarity: The paper is clear and flows very well from motivation and background to theory and experimentation.

Relation to Prior Work: No significant comments on this part.

Reproducibility: Yes

Additional Feedback: - Although it is not mentioned in the paper, sharing parameters would also allow batching the evaluation of certain layers with a GPU across tasks, which speeds up inference time. - I did not understand Fig 2c,d; adding a small caption for those parts would help. - I think the last word "for" in line 69 is a typo - In Fig 2, the columns should probably be sorted by appearance in the network, not alphabetically - In the last experiment, I don't fully understand why 1-step MAML doesn't have enough memory but sigma-Reptile (which also requires gradients w.r.t. \phi,\sigma) has enough memory. - Appendix C was useful in understanding the differences between meta-learning learning rate and shrinkage ===================== POST REBUTTAL I'm happy with the answers, I like the paper, I've increased my score from 7 to 8. Congrats on the nice work :)


Review 2

Summary and Contributions: The authors propose a bayesian approach to meta-learning. More specifically they make use of bayesian shrinkage to modularize a neural network architecture into large and generic modules that can be reused as-is on new tasks and smaller modules that can be adapted to new tasks with a much lower risk of overfitting. The model is well justified, each step is described in details and the performance of the approach is empirically validated by experiments on both synthetic and real-world tasks across several domains.

Strengths: The paper is well written, the proposed method theoretically sound and thoroughly validated by a large set of experiments. To my knowledge, this is the first work using bayesian meta-learning to automatically choose which parts of the network should be finetuned on the new tasks. This work is interesting and of very high quality, the in-depth discussion of each contribution and detailed experimental protocol will make it easy for the community to reuse and extend the proposed techniques.

Weaknesses: The main weakness of this work is the limited performance improvement on standard tasks. In my opinion, this is not a huge issue since the authors clearly state that the focus of this work is long term adaptation, which is challenging and not required in most of the standard benchmarks.

Correctness: The claims are discussed in-depth, theoretically justified when needed, and empirically validated by an extensive set of experiments. These experiments clearly demonstrate that the approach works in the targeted setting (long term adaptation). Supplementary materials also contain experiments showing the behavior of the method in different settings on the standard tasks used in meta-learning.

Clarity: The paper is very well written and pleasing to read.

Relation to Prior Work: The authors position their work in the meta-learning landscape, clearly discussing its relation to the other approaches existing in the literature.

Reproducibility: Yes

Additional Feedback: As stated above, I think this paper is of very high quality and is a valuable contribution opening the path for future research in the meta-learning domain. Update: I am satisfied with the author's response and keep my evaluation unchanged.


Review 3

Summary and Contributions: This paper introduces generalizations of MAML, iMAML and Reptile, where the modular structure of the parameters can be leveraged to control how much certain modules need to be adapted and which module are task-agnostic.

Strengths: I really liked the inspiration from the Bayesian literature to use shrinkage to have a modular structure emerge from meta-learning. The fact that learning this modular structure leads to conclusions which are not necessarily aligned with (Raghu et al., 2019) is very interesting. I also really liked the non-standard application of meta-learning to TTS. I think it is great that this paper goes beyond the usual few-shot image classification problems and tackle real-world problems, and I am looking forward to more progress in this direction. (Raghu et al., 2019) Aniruddh Raghu, Maithra Raghu, Samy Bengio, and Oriol Vinyals. Rapid learning or feature reuse? Towards understanding the effectiveness of MAML

Weaknesses: A. Major concerns 1. Can you comment on the choice of a Normal prior for your shrinkage variants, as opposed to a sparsity inducing prior, such as a Laplace or a Spike and Slab prior? Sparsity inducing priors would probably be more intuitive for a better modularity (where some layers would require no adaptation at all, as opposed to a small adaptation). 2. The paper is motivated with the benefit of having a more modular approach to meta-learning. The experiments do show that the sigma version of the different algorithms learn different scales of adaptation. However there is no experiment showing the benefit of these approaches for some of the aspects that motivated this approach (interpretability, causality, transfer learning or domain adaptation), beyond the standard performance in the few-shot learning setting. B. Moderate concerns 1. Lines 27-28: "As data increases, these hard-coded modules may become a bottleneck for further improvement.". In all the experiments of this paper, we are in the few-shot learning setting. This undermines a bit the argument that hard-coded modules is a bottleneck in the limit of large data (also in line 2. Line 104: "When L is large, as in TTS systems, this becomes computationally prohibitive.". There should be a mention here as to why TTS systems require large L. Why can't we meta-learn a TTS system with small L? (also lines 21-23: "few-shot text-to-speech synthesis (TTS), characterized by [...] long adaptation horizons"). 3. There was no code released as part of the Supplementary Material. However, I appreciated the detailed hyperparameters reported in the Supplementary Material. 4. It would be more fair to compare this work against methods that learn their learning rate, such as meta-SGD (Li et al., 2017, [30] in the submission) or meta-Curvature (Park et al., 2019 [15] in the submission). There is a mention of the difference between these prior works and this submission in the "Related Work" section (lines 74-76), and while I understand how the two are orthogonal, the effect on the adaptation might be similar, and are worth comparing. 5. Lines 287-288: "the MOS scores indicate that shrinkage may improve out-of-domain generalization.". How is this out-of-distribution generalization? C. Minor concerns 1. Line 250: "Most \sigma^{2} values are too small to be visible.". Using a log-scale for \sigma^{2} might help. Currently Figures 2a, 2b, and 3 are not really helpful quantitatively. (Li et al., 2017) Zhenguo Li, Fengwei Zhou, Fei Chen, Hang Li, Meta-SGD: Learning to Learn Quickly for Few-Shot Learning (Park et al., 2019) Eunbyung Park, Junier B. Oliva, Meta-Curvature

Correctness: The theoretical results are correct. The empirical methodology is also correct, and the extensive description of the hyperparameters used in the Appendix are very appreciated.

Clarity: The paper is very clearly written. Although I haven't read the Supplementary Materials in details (only the sections mentioned in the main text), it looks very comprehensive.

Relation to Prior Work: The relation to prior work is clear, especially work that uses a modular approach to meta-learning, and work that learn a preconditioning on the gradient update, which are most relevant to this submission.

Reproducibility: Yes

Additional Feedback: I am leaning towards increasing my score if the concerns raised above are addressed. After authors response -------------------------- After reading the authors response, I feel like my comments were all addressed. I am happy to increase my score to 7.


Review 4

Summary and Contributions: The paper proposes a meta learning approach based on Bayes shrinkage estimation. The proposed approach is developed to automatically splitting the model to shared modules and task-specific modules.

Strengths: + The idea to discover sharing modules is interesting. The approach could be extended to other fields. + The improvement looks large comparing with baseline methods + The author proposed a general framework to include previous works (table 1), which could give a better understanding of the relationship between meta-learning approaches. + The experiment is clear to support the proposed approach.

Weaknesses: - No code is provided. Codes could help readers know how the approach is implemented. - In table 4, appendix d, why the learning rate for different approaches is different for different approaches (also in table 5, 7, 8, 9)? The sigma-MAML uses learning rate 6.3e-3, the sigma-Reptile is 6.2e-3, how much benefits could bring by 0.1e-3? Does this mean that the proposed method is not robust, which needs tuning the learning rate very carefully to gain a good performance? - The experiment is limited in small scale networks (e.g. network mentioned in appendix e.1), is the proposed approach also effective in modern CNN architectures?

Correctness: yes

Clarity: The paper is very clear.

Relation to Prior Work: yes

Reproducibility: Yes

Additional Feedback: The hyper-parameters are provided in detail in the appendix, but no code is provided. Fig. 2 (a) and (b), the learned \sigma^2 is too small in bn0 to conv 3 layers. The author could consider using log-scale y-axis to plot the bins.

[Author Response · NeurIPS 2020]

Thank you all for the encouraging comments and helpful suggestions. Due to the space limit, we only respond to main
concerns, but will incorporate all comments in the final revision. Citations in this rebuttal refer to those in the main paper.

**Reviewer 1**
- **Bold text**: We bold the highest mean accuracy in all tables and show the bar of our model ($\sigma$-reptile) for readability
in Fig. 5. We will instead bold the statistically best models in the revision as you suggest.
- **When do we need long adaptation**: We will expand this in the revision. Generally speaking, long adaptation allows
higher model capacity for optimization-based meta learning [1-3]. The augmented Omniglot dataset was designed to
be challenging for short-horizon optimization, and MAML was found previously not to work for few-shot TTS [6].
- **Selecting modules according to $\sigma$ and setting to 0 in testing**: This is a very good question. Bayesian approaches
always allow some probability of selecting any module. It is common practice in the literature (e.g., Bayesian /
regularization-based feature selection [40] and model pruning) to choose features (modules in our case) with large
coefficients ($\sigma^2$). As explained in Sec D.2 we apply a weak regularizer on $\sigma^2$ so that the learned value for task
independent modules is close to zero. In practice, we often find that $\sigma^2$ of most modules is numerically indistinguishable
from 0, which reduces the gap between meta-training and meta-testing (where we set $\sigma = 0$). One exception is the
TTS experiment where some pruned modules have non-zero $\sigma^2$. All res-blocks have equal size in that case. There, we
choose the threshold with the best trade-off between pruned model size and validation performance. Your comment on
the relationship of $\sigma^2$ with the module size is however worth further investigation. We will add a discussion of this issue.
- **BatchNorm**: Each set of batch-norm parameters (offset, scale) has their own $\sigma$. Running stats are not parameters
but are computed from inputs. For consistency, all algorithms use transductive learning as in MAML and Reptile.
- **1-step MAML vs $\sigma$-Reptile**: MAML requires computing second-order gradients even with 1-inner step.

**Reviewer 2** We appreciate your assessment on the strengths of our work.

**Reviewer 3**
- **Choice of a Normal prior vs sparsity inducing prior**: The shrinkage prior is sparsity inducing for task parameters.
Given a weak prior on $\sigma^2$, the marginal prior distribution of task parameters is heavy-tailed with tied variance in a
group. Similarly, Laplace (spike-and-slab) priors is equivalent to a normal random variable with an exponential (mix
of point masses) prior on the variance. The estimate of $\sigma$ will shrink towards zero if the data does not support task
parameters deviating from the prior mean, and thus lead to sparse selection of parameter groups.
- **Showing benefit for some of the aspects that motivated this approach (interpretability, causality, transfer learn-
ing or domain adaptation)**: The cited line 30 is a general statement for the purpose of learning resuable modules in
the literature. We will clarify that we do not evaluate on all of these aspects. One of our main motivations is for saving
space in deployment by sharing the majority of parameters across tasks, which we demonstrate in all experiments.
Reviewer 1 also suggests that "*sharing parameters would also allow batching the evaluation of certain layers with
a GPU across tasks, which speeds up inference time*". Finally, the discovered middle layers in the WaveNet stack help
us better interpret the learned features in the different layers.
- **Bottleneck of hard-coded modules in few-shot domains**: This bottleneck may still exist in few-shot domains. Take
TTS for example, training a model from scratch typically requires at least 10K utterances. Learning a model with
$\sim 100$ utterances is a few-shot learning task. Nevertheless, the performance of recent works [5-7] with a hard-coded
task-specific component becomes saturated after around 10 utterances. Simply increasing the layer size of the task
specific component does not fix the problem.
- **Requiring large number of adaptation steps ($L$) in TTS**: We will clarify the reference to [6] of this requirement in
TTS. Generally speaking, allowing more adaptation steps increases the capacity of optimization-based methods [1-3].
- **Comparison to methods that learn learning rates**: We discuss the relationship of our method with those approaches
in Related Work and Appendix C. In Appendix C, we also compare with meta-SGD on a small synthetic example and
demonstrate the different behaviors for long adaptation. For real-data experiments, meta-SGD/-Curvature require back-
propagation through gradients and thus do not scale to long-adaptation regimes. Finally, SGD performs much worse than
Adam for the WaveNet model. It is not clear how to combine meta-SGD/-Curvature with Adam for a fair comparison.
- **Out-of-domain generalization in TTS**: We refer to the different size of the task dataset in meta-training and testing.
We will use a more accurate word in the revision.

**Reviewer 4**
- **Different learning rate in different approaches**: We run a random hyper-parameter search with 100 seeds for each
algorithm in each experiment and use the best found value from the validation task set. This is to ensure a fair comparison.
Particularly, we find that the reported hyper-parameters in the original papers (MAML, iMAML, Reptile) are not
optimal in our implementation. This small difference in learning rate makes essentially no difference in performance.
- **Small scale networks** The WaveNet model in the TTS experiment is a large network with 30 residual blocks. Every
block is a dilated and gated causal-convolutional net (see architecture in Figure 12 of Appendix E.2). There are 3M
parameters in total. In contrast to most other few-shot meta-learning works that use a pretrained residual network as
backbone and meta-learn only the last few layers, we meta-learn the entire CNN stack.

[Meta-Review · NeurIPS 2020]

This paper presents a modular meta-learning method where Bayesian shrinkage is used to decide which modules to be updated for a new task. All reviewers agree that the modular approach has a clear motivation, is theoretically sound, and is thoroughly validated by a large set of experiments.